# RPLP1 restricts HIV-1 transcription by disrupting C/EBPβ binding to the LTR

Weijing Yang[1,2,3], Hong Wang[1,2,3], Zhaolong Li[1,2,3], Lihua Zhang [4], Jianhui Liu[4], Frank Kirchhoff [5], Chen Huan [1,2,3] ✉ & Wenyan Zhang [1,2,3] ✉

Long-term non-progressors (LTNPs) of HIV-1 infection may provide important insights into mechanisms involved in viral control and pathogenesis. Here, our results suggest that the ribosomal protein lateral stalk subunit P1 (RPLP1) is expressed at higher levels in LTNPs compared to regular progressors (RPs). Functionally, RPLP1 inhibits transcription of clade B HIV-1 strains by occupying the C/EBPβ binding sites in the viral long terminal repeat (LTR). This interaction requires the α-helixes 2 and 4 domains of RPLP1 and is evaded by HIV-1 group M subtype C and group N, O and P strains that do not require C/EBPβ for transcription. We further demonstrate that HIV-1-induced translocation of RPLP1 from the cytoplasm to the nucleus is essential for antiviral activity. Finally, knock-down of RPLP1 promotes reactivation of latent HIV-1 proviruses. Thus, RPLP1 may play a role in the maintenance of HIV-1 latency and resistance to RPLP1 restriction may contribute to the effective spread of clade C HIV-1 strains.

Long-term non-progressors (LTNPs) are rare study participants living with HIV-1 who remain asymptomatic with normal and stable CD4 + T cell counts for more than 10 years after diagnosis in the absence of combination antiretroviral therapy (cART)[1–3]. Discovering the mechanisms underlying non-progressive HIV-1 infection may help to improve therapeutic and curative strategies. Host genetics and immune response factors, such as specific HLA alleles and sequences polymorphisms in the IL28B or CXCR6 genes, have been reported to play roles in the effective control of HIV-1 replication and disease progression[4–6]. Viral features including defective viral accessory genes and variations in the Envelope glycoprotein have also been implicated in non-progressive infection[7–11]. The HUSH complex (TASOR, MPP8, and periphilin) has been reported to silence proviral transcription in CD4+ memory T cells and to help HIV-1 to escape eradication by the host immune system[12,13]. Altogether, however, the mechanisms by which host factors control HIV-1 replication and prevent disease progression are still largely unclear. Host restriction factors, such as TRIM5α, APOBEC3G (A3G), Tetherin, SAMHD1, SERINC5, GBP5 and IFI16, play key roles in the first line of defense against HIV-1[14–21]. Here, we performed mass spectrometry profiling of cells derived from LTNPs or regular progressors (RPs) to identify as-yet-unknown antiviral host factors suppressing HIV-1 replication and promoting non-progressive infection. Our approach revealed that ribosomal protein lateral stalk subunit P1 (RPLP1) is expressed at higher levels in LTNPs compared to RPs, suggesting a relevant role in HIV-1 inhibition and silencing in LTNPs.

RPLP1 interacts with RPLP0 and RPLP2 to form the pentameric ribosomal stalk complex (also called a pentameric P complex), which plays an essential role in translation[22,23]. The P complex has been extensively investigated and associated with several pathological conditions such as autoimmune diseases, human cancer, and viral infections[24–29], but the biological functions of individual proteins are largely unknown. An in vivo knockout model of the RPLP1 protein showed that overall protein synthesis is not affected[30], suggesting that RPLP proteins might have extra-ribosomal functions.

[1]Department of Infectious Diseases, Infectious Diseases and Pathogen Biology Center, The First Hospital of Jilin University, Changchun, China. [2]Institute of Virology and AIDS Research, The First Hospital of Jilin University, Changchun, China. [3]Key Laboratory of Organ Regeneration and Transplantation of The Ministry of Education, The First Hospital of Jilin University, Changchun, China. [4]State Key Laboratory of Medical Proteomics, Dalian Institute of Chemical Physics, Chinese Academy of Science, Dalian, China. [5]Institute of Molecular Virology, Ulm University Medical Center, 89081 Ulm, Germany. ✉e-mail: lmsxsml@jlu.edu.cn; zhangwenyan@jlu.edu.cn

In this study, we show that RPLP1, which is expressed at higher levels in LTNPs compared to RPs, inhibits the transcription of HIV-1 group M subtype B strains that require the transcriptional factor C/EBPβ but not of clade C HIV-1 strains that currently dominate the AIDS pandemic, as well as groups N, O and P. Chromatin immunoprecipitation (ChIP) and DNA binding assays revealed that RPLP1 binds to the C/EBPβ binding sites in the LTR thereby suppressing C/EBPβ binding and consequently viral gene expression. The translocation of RPLP1 from cytoplasm to the nucleus, induced by HIV-1 infection, represents a crucial prerequisite for its anti-HIV-1 activity. Identification of RPLP1 as a transcriptional repressor of HIV-1 subtype B strains provides a target to modulate reactivation of the latent viral reservoirs in cure strategies.

## Results

### RPLP1 inhibits HIV-1 replication

We identified three LTNPs controlled viremia to below levels of detection of the most sensitive polymerase chain reaction (PCR) assay for more than 17 years and lack of other viral infections. To identify potential host factors promoting HIV-1 control and non-progressive infection, we employed mass spectrometry-based proteomic profiling to analyze these three LTNPs and two PRs, defined as HIV-1 infected individuals who developed AIDS and received cART (Supplementary

Table 1), then identified several proteins expressed at higher levels (>2-fold) in LTNPs compared to RPs (Fig. 1a). Among them, RPLP1 was the most markedly up-regulated gene in LTNPs with the highest fold change value of 11.38, hinting that RPLP1 may play a role in host suppression of HIV-1. Thus, we subsequently focused on RPLP1 to gain functional insight into its anti-HIV-1 activity. We generated stable over-expression and knock-down RPLP1 Jurkat T cells and THP-1 monocyte-derived macrophages and infected them with HIV-1 pNL4-3-deltaE-EGFP pseudo-typed with the VSV-G protein produced from HEK293T cells. Overexpression of RPLP1 in Jurkat and THP1 cells reduced the susceptibility of the cells to HIV-1 infection by ~40% and ~60%, respectively, while silencing of RPLP1 expression enhanced it by ~2-fold (Fig. 1b, c). Similarly, stable over-expression of RPLP1 in MT-4 cells showed lower susceptibility to HIV-1 infection and lower mRNA levels of viral *rev, vpu, vif,* and *gag* genes, while knock-down of RPLP1 expression by ~50% significantly enhanced susceptibility to HIV-1 infection and higher mRNA levels (Supplementary Fig. 1a, b). Notably, the Mean Fluorene Intensity (MFI) of GFP consistently supports an inverse correlation between RPLP1 levels and HIV-1 replication (Fig. 1b, c and Supplementary Fig. 1a). Since RPLP1 was reported to function in translation, we conducted a CCK8 assay to compare the viability of cells overexpressing or silenced RPLP1. The results revealed no significant difference in cell viability between control and RPLP1

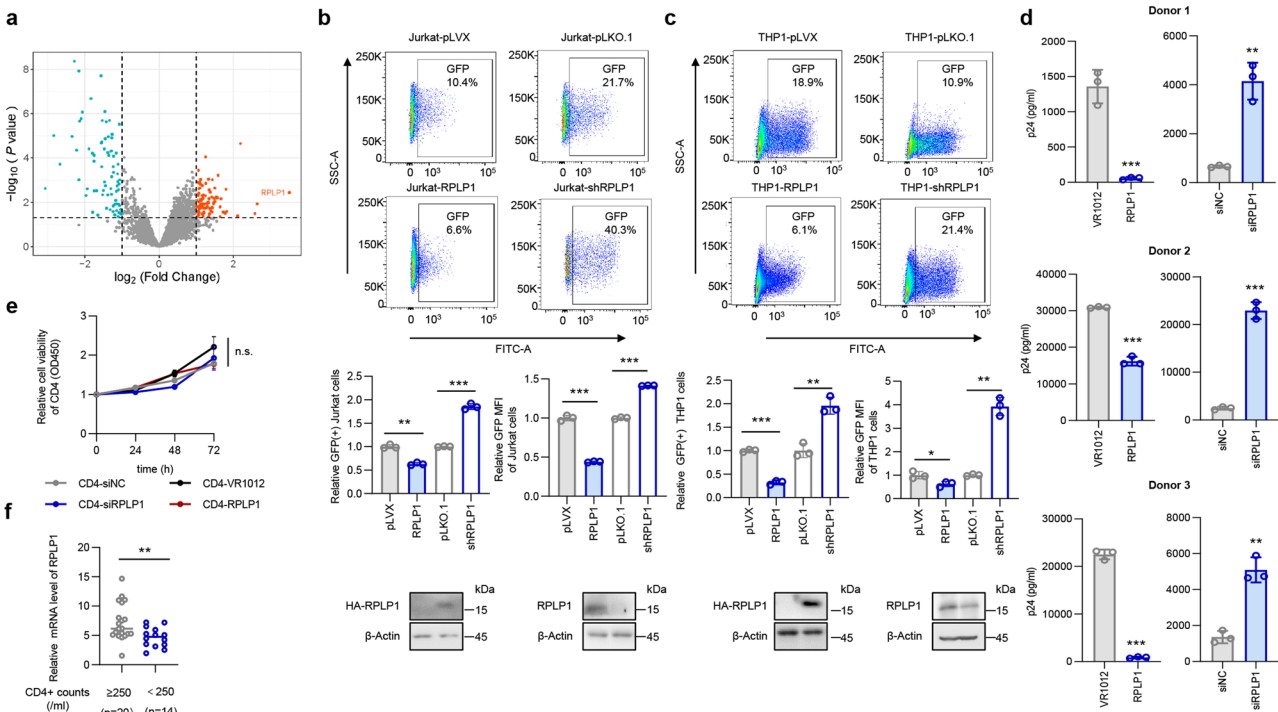

**Fig. 1 | RPLP1 inhibits HIV-1 replication. a** Volcano plots showing different expressed proteins in LTNPs (*n* = 3 donors) vs RPs (*n* = 2 donors). Upregulated proteins in LTNPs are shown in red, downregulated in green (fold change >2, *P* < 0.05). *P* values are calculated by limma package (version 3.58.1) in which a linear model is fit, then moderated *t*-statistics are calculated by empirical Bayes moderation of the standard errors towards a global value. Dashed lines indicate *P* value of 0.05 and 2-fold change on Y and X axis, respectively. **b** RPLP1 inhibits HIV-1 replication in Jurkat cells. Jurkat cells with altered RPLP1 levels (lower) were infected with HIV-1 NL4-3-EGFP virus and the percentage of GFP-positive cells was measured by flow cytometry (upper). Ratio of GFP-positive cells and relative mean fluorescence intensity (MFI) normalized to control cells were calculated (middle). **c** RPLP1 inhibits HIV-1 replication in THP-1 monocyte-derived macrophages. THP1 cells with altered RPLP1 levels (lower) were treated with PMA (100 nM) to differentiate into macrophages, then infected with HIV-1 NL4-3-EGFP virus. GFP-positive cells were measured by flow cytometry (upper). Ratio of GFP-positive cells and MFI

normalized to control cells were calculated (middle). **d** RPLP1 inhibits HIV-1 replication in primary CD4+ T cells. CD4+ T cells from HIV-negative participants (*n* = 3 donors) were transfected with HA-RPLP1 or siRNA targeting RPLP1, infected with HIV-1 NL4-3. HIV-1 yield in supernatant was measured with p24 ELISA assay. **e** Alternation of RPLP1 didn't affect viability of primary CD4+ T cells. **f** RPLP1 level was negatively correlated with disease progression of participants living with HIV-1. Participants with HIV-1 were split into two groups based on CD4+ cell counts/ml (≥250, *n* = 20 donors; <250, *n* = 14 donors), and RPLP1 mRNA levels were compared, with dots represent independent donors. Immunoblots in (**b, c**) are representative of three independent experiments. Quantification in (**b–e**) was shown as means ± SDs from three independent experiments. *P* values were calculated by the two-tailed Student's *t* test (**b–f**). *\**P* < 0.05, \*\**P* < 0.01, \*\*\**P* < 0.001, n.s. denotes no significance. See also Supplementary Fig. 1. Source data are provided as a Source Data file.

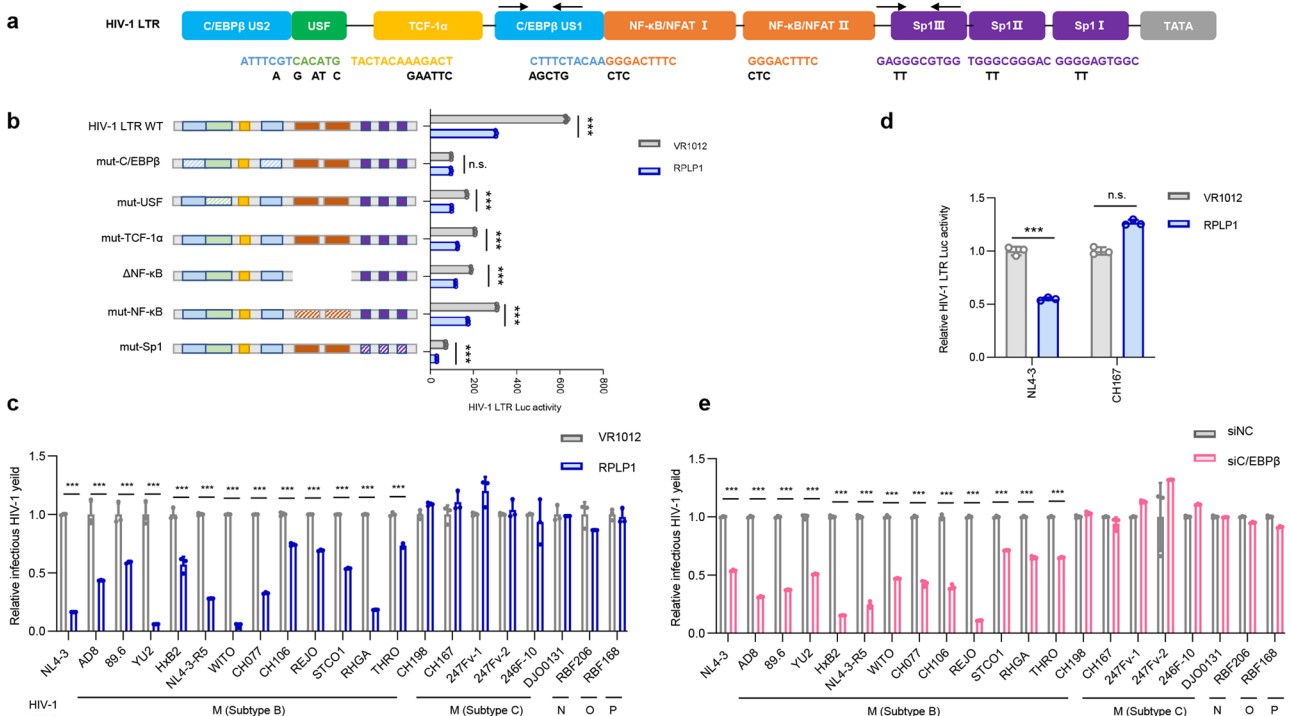

**Fig. 2 | RPLP1 inhibits HIV-1 replication by suppressing viral LTR activity.**
**a** Schematic of HIV-1 LTR mutants. HIV-1 LTR luciferase reporter constructs were generated with mutations in various transcription factor binding sites. Primers used in the ChIP assays were indicated. **b** C/EBPβ binding motif is essential for the RPLP1-mediated HIV-1 transcription inhibition. HEK293T cells were transfected with wild-type or mutant HIV-1 LTR-luciferase reporter, pRenilla plasmid, and either control vector or Flag-RPLP1. After 48 h, LTR activity was measured using a dual-luciferase reporter assay, with the activity of cells transfected with wild-type LTR set as 1. **c** RPLP1 inhibits HIV-1 production of subtype B, but not subtype C or group N, O, P. Various HIV-1 constructs were transfected into HEK293T cells with or without Flag-RPLP1. Infectious virus yield was quantified using the TZM-bl reporter cell infectivity assay, and the yield of virus from cells transfected with HIV-1 construct plus

control vector was set as 1. **d** RPLP1 suppresses HIV-1 LTR activity of subtype B (NL4-3), but not subtype C (CH167). HEK293T cells were transfected with indicated constructs. After 48 h, LTR activity was measured, and the luciferase activity of cells transfected with HIV-1 LTR-luciferase reporter, pRenilla plasmid, and control vector set as 1. **e** C/EBPβ silencing significantly decreased viral production of HIV-1 subtype B but not subtype C or group N, O, P. Control or C/EBPβ silencing HEK293T cells were transfected with various HIV-1 constructs and the resulting virus yield was measured using the TZM-bl reporter cell infectivity assay, relate to the control cells transfected with HIV-1 construct. Data in (**b**–**e**) was shown as means ± SDs from three independent experiments. *P* values were calculated by the two-tailed Student's *t* test (**b**–**e**). ***$P < 0.001$, n.s. denotes no significance. See also Supplementary Fig. 2. Source data are provided as a Source Data file.

knockdown or overexpressing cells, indicating that alterations in RPLP1 levels have minimal influence on cell viability (Supplementary Fig. 1d). The antiviral effect of RPLP1 was further verified in CD4+ T cells derived from HIV-negative study participants ($n$ = 3, Fig. 1d) after nucleofection with HA-RPLP1 or siRNA against endogenous RPLP1 (Supplementary Fig. 1c) without cytotoxic effects (Fig. 1e). In order to assess the relationship between RPLP1 and the disease progression after HIV-1 infection, we further detected the mRNA level of RPLP1 in CD4+ T cells isolated from study participants living with HIV-1. Compared to participants with lower CD4+ cell counts, the mRNA level of RPLP1 in participants with higher CD4+ cell counts were higher, indicating that there was a negative correlation between RPLP1 expression and disease progression (Fig. 1f). Additionally, transfection of HEK293T cells with an HIV-1 NL4-3 proviral construct and increasing amounts of a Flag-RPLP1 expression vector also confirmed dose-dependent inhibition of HIV-1 by TZM-bl infection assay (Supplementary Fig. 1e, f). Collectively, these data demonstrated that RPLP1 inhibits HIV-1.

### RPLP1 inhibits HIV-1 replication by suppressing viral LTR activity
Having confirmed the inhibitory effect of RPLP1 on HIV-1 in three different cell lines and primary T cells, we then examined which step in the HIV-1 replication cycle is affected by RPLP1. We quantified late-RT, 2-LTR and two-step Alu PCR in control or RPLP1 overexpressing MT-4 cells infected with HIV-1, which represent the HIV-1 reverse transcription, nuclear entry and integration, respectively[31,32]. The

results showed that RPLP1 over-expression had no effect on proviral DNA levels (Supplementary Fig. 2a), suggesting that RPLP1 might suppress HIV-1 transcription. Thus, we measured the impact of RPLP1 on HIV-1 LTR promoter activity and observed that increasing amounts of RPLP1 significantly suppressed HIV-1 LTR activity both in the absence and presence of Tat (Supplementary Fig. 2b), while the expression of Tat driven by a CMV promoter was not affected (Supplementary Fig. 2c). These results indicate that RPLP1 inhibits LTR-driven HIV-1 transcription in a Tat-independent manner.

### C/EBPβ binding sites in the LTR are required for RPLP1 antiviral activity
The HIV-1 LTR comprises numerous transcription factor interaction sites to regulate viral transcription[33]. In order to determine which elements in HIV-1 LTR are targeted by RPLP1, we utilized HIV-1 LTR luciferase plasmids containing mutations in their C/EBPβ, USF, TCF-1α, NF-κB/NFAT and Sp1 transcription factor binding sites[34] (Fig. 2a). Only the mutant LTR lacking the C/EBPβ interaction sites was not affected by RPLP1 expression (Fig. 2b and Supplementary Fig. 2d), indicating that C/EBPβ binding sites in the LTR are required for RPLP1 antiviral activity.

Transcription factor C/EBPβ was initially identified as a member of a C/EBP family, also called NF-IL6[35]. C/EBPβ has been shown to contribute to the enhancement of HIV-1 transcription through binding to its C/EBPβ binding sites, in particular in the monocyte-macrophage lineage[36–39]. To assess the antiviral spectrum and requirement of C/EBPβ binding sites for the inhibitory effect of RPLP1, we tested a panel

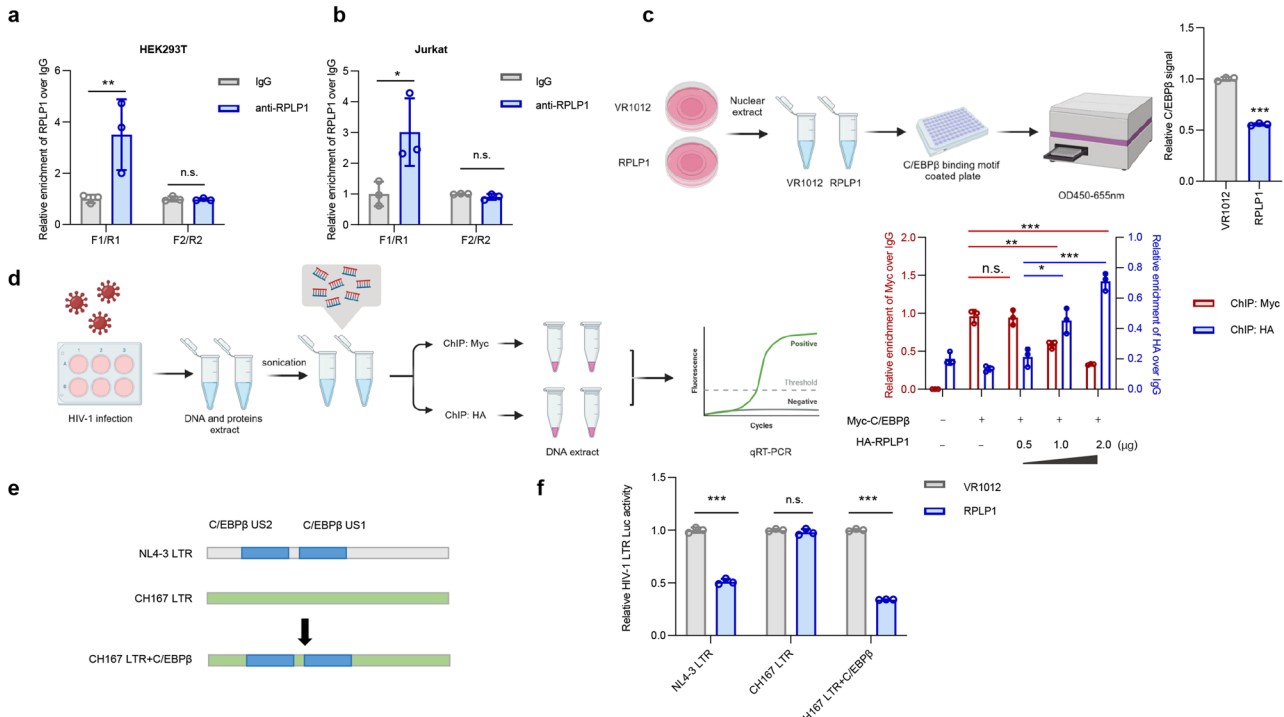

**Fig. 3 | RPLP1 competes with the transcription factor C/EBPβ to bind HIV-1 LTR.** **a**, **b** RPLP1 is recruited to the HIV-1 NL4-3 LTR via the C/EBPβ binding sites. Chromatin from HIV-1-infected HEK293T (**a**) or Jurkat (**b**) cells was immunoprecipitated with anti-RPLP1 antibody or the negative control IgG and analyzed by RT-qPCR with the indicated primers spanning C/EBPβ (F1/R1) or Sp1 (F2/R2) binding sites in the LTR. **c**, **d** C/EBPβ recruitment to the LTR promoter is disrupted by RPLP1. **c** HEK293T cells were transfected with control or Flag-RPLP1 construct. 48 h later, cells were harvested and nuclear proteins were isolated. Normalized amount of nuclear extracts were subjected for the TransAM C/EBPβ binding assay (Active-Motif). **d** HEK293T cells co-transfected with Myc-C/EBPβ plus increasing doses of HA-RPLP1 were infected with HIV-1 NL4-3 and split into two parts, followed by separately subjected to ChIP assays using antibody against HA or Myc with IgG as

the negative control, respectively. RT-qPCR was performed with the primers spanning the C/EBPβ binding site. **e** Schematic of CH167 LTR chimeric mutant. LTR-luciferase reporter construct of HIV-1 CH167 was modified by inserting two C/EBPβ binding sites. **f** RPLP1 inhibits LTR activity of CH167 harboring C/EBPβ binding sites. HEK293T cells were transfected with indicated constructs, and the cells were collected at 48 h post transfection. The LTR activity was detected with the activity of cells transfected without RPLP1 set as 1. Quantifications in (**a**–**d**, **f**) are shown as means ± SDs from three independent experiments. *P* values were calculated by the two-tailed Student's *t* test (**a**–**d**, **f**). *$P < 0.05$, **$P < 0.01$, ***$P < 0.001$, n.s. denotes no significance. The schematic workflow in (**c**, **d**) is generated using BioRender (http://biorender.com/). See also Supplementary Fig. 3. Source data are provided as a Source Data file.

of 21 infectious molecular clones (IMCs) of HIV-1[40–42]. A variety of subtype B and C HIV-1 constructs and individual group N, O, and P IMCs were transfected into HEK293T cells together with a construct expressing Flag-RPLP1 or an empty control vector. Two days later, the yield of infectious HIV-1 was quantified by TZM-bl infection assay. We found that RPLP1 significantly inhibited the yield of all 13 HIV-1 subtype B IMCs albeit to varying degrees, while infectious virus production of HIV-1 subtype C and non-M group IMCs was not affected (Fig. 2c). Notably, C/EBPβ binding sites are conserved in the LTRs of subtype B but not subtype C and group N, O and P viruses (Supplementary Fig. 2e). Accordingly, the LTR activity of subtype B (NL4-3) but not subtype C (CH167) was significantly suppressed by RPLP1 (Fig. 2d). Knock-down of endogenous C/EBPβ with siRNA (Supplementary Fig. 2f) decreased the production HIV-1 subtype B clones harboring C/EBPβ binding sites, while the production HIV-1 subtype C and group N, O and P clones remained unaffected (Fig. 2e). In addition, over-expression of C/EBPβ promoted the production and LTR activity of NL4-3, but not CH167 (Supplementary Fig. 2g, h). Altogether, these results show that the antiviral effect of RPLP1 depends on C/EBPβ binding sites that are found in HIV-1 clade B LTRs but absent in LTRs of subtype C and group N, O and P strains.

## RPLP1 competes with the transcription factor C/EBPβ for HIV-1 LTR binding

Since HIV-1 inhibition by RPLP1 was dependent on C/EBPβ binding sites in the LTR, we further investigated whether RPLP1 specifically interacts

with the C/EBPβ binding sites. Chromatin immunoprecipitation (ChIP) assays showed significant enrichment of binding by an anti-RPLP1 antibody to the fragment containing a C/EBPβ binding site in both HEK293T and Jurkat cells, whereas there was no enrichment with the control fragment containing an Sp1 interaction site upon HIV-1 infection (Fig. 3a, b). In line with our finding that the production of clade C HIV-1 strains is not affected by RPLP1, no enrichment of RPLP1 was observed for the clade C CH167 LTR (Supplementary Fig. 3a). These results further supported that RPLP1 interacts with C/EBPβ sites in HIV-1 LTRs and led us to explore the impact of RPLP1 on LTR-C/EBPβ interaction. Over-expression of C/EBPβ enhanced HIV-1 NL4-3 yield, whereas RPLP1 reduced it (Supplementary Fig. 3b, c). Similarly, RPLP1 overexpression suppressed the enhancing effect of C/EBPβ on HIV-1 LTR activity (Supplementary Fig. 3d) and significantly reduced binding of C/EBPβ to the HIV-1 LTR (Supplementary Fig. 3e). ELISA assays for DNA binding confirmed that RPLP1 reduces the amount of C/EBPβ bound to its target DNA sequence (Fig. 3c). To further determine whether RPLP1 competes with C/EBPβ for HIV-1 LTR binding, cells co-transfected with C/EBPβ-Myc plus increasing doses of HA-RPLP1 were infected with HIV-1 and split into two parts to separately measured with ChIP assays, with antibodies against Myc or HA, respectively (Fig. 3d). These analyses confirmed that increasing binding of RPLP1 (Fig. 3d, blue bars) is associated with reduced binding of C/EBPβ to the HIV-1 LTR (Fig. 3d, red bars). Finally, introduction of C/EBPβ binding sites into the clade C CH167 LTR (Fig. 3e) rendered it susceptible to the inhibitory effect of RPLP1 (Fig. 3f and Supplementary Fig. 3f).

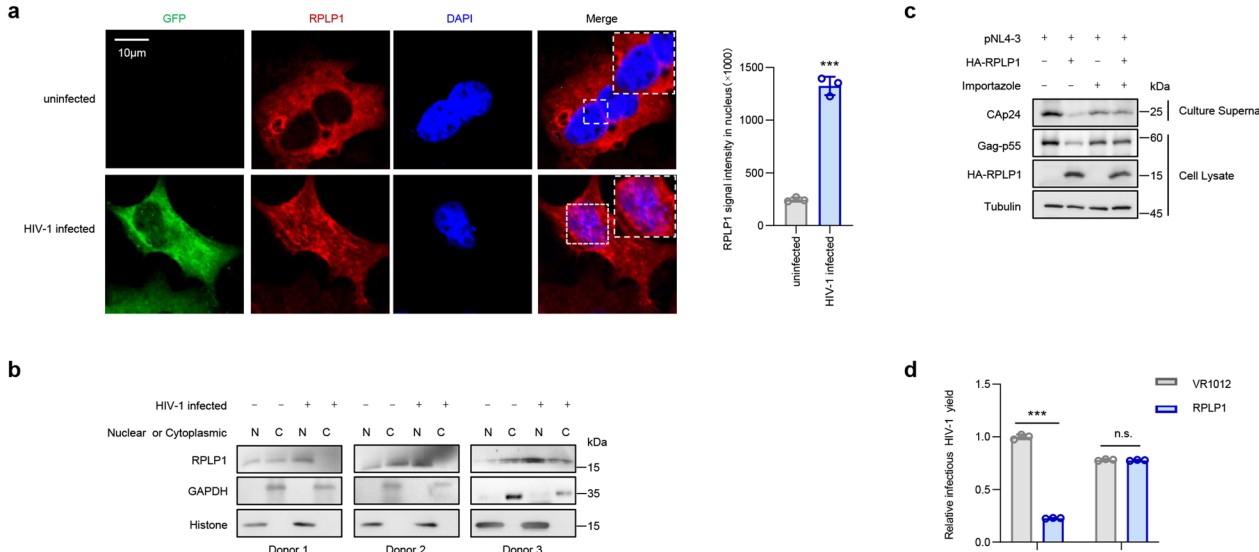

**Fig. 4 | HIV-1 infection induces cytoplasm-to-nucleus translocation of RPLP1.**
**a** HIV-1 infection induced cytoplasm-to-nucleus translocation of RPLP1 in HeLa cells. HeLa cells infected with HIV-1 NL4-3-EGFP virus for 48 h were detected by immunofluorescence assay, and images were captured with a Zeiss LZM710 confocal microscope. Scale bars, 10 μm (left). The fluorescence intensity of RPLP1 in nucleus was analyzed using ImageJ (right). **b** HIV-1 infection induced cytoplasm-to-nucleus translocation of RPLP1 in CD4+ T cells. CD4+ T cells isolated from HIV-negative study participants (*n* = 3 donors) were infected with NL4-3 virus for 48 h, and then the nuclear-cytoplasm separation assay was performed. N, nuclear; C, cytoplasm. **c, d** The nuclear import inhibitor Importazole impaired the anti-HIV-1 function of RPLP1. HEK293T cells were co-transfected with pNL4-3 with or without HA-RPLP1. Thirty-six hours later, the cells were treated with Importazole (40 μM) for additional 12 h. Then, the cells were harvested and examined with immunoblotting (**c**), and the viral yield in culture supernatant was detected with TZM-bl assays (**d**). Immunofluorescence images in (**a**) and immunoblots in (**b** and **c**) are representative of three independent experiments. Quantifications in (**a** and **d**) are shown as means ± SDs from three independent experiments. *P* values were calculated by the two-tailed Student's *t* test (**a, d**). ***P < 0.001, n.s. denotes no significance. See also Supplementary Fig. 4. Source data are provided as a Source Data file.

Additionally, we manipulated RPLP1 levels in CD4+ cells derived from HIV-negative study participants as in Fig. 1d, and assessed its influence on HIV-1 subtype C infection. The results demonstrated that the replication of subtype C, which is independent of C/EBPβ, is not modulated by RPLP1 in primary CD4+ T cells (Supplementary Fig. 3g). These data show that RPLP1 inhibits HIV-1 subtype B transcription by competing for C/EBPβ binding to the LTR promoter.

## HIV-1 infection induces cytoplasm-to-nucleus translocation of RPLP1

RPLP1 is usually found in pentameric P complexes as well as in free form in the cytoplasm of the cell[23,24], while proviral transcription of HIV-1 occurs in the nucleus. To further elucidate the inhibitory mechanism, we investigated where RPLP1 is located during HIV-1 clade B infection. We infected HeLa, THP1, as well as HIV-negative study participants-derived CD4+ T cells and found that, compared with uninfected control cells, HIV-1 NL4-3 infection induced significant cytoplasm-to-nucleus translocation of RPLP1 by confocal microscopy (Fig. 4a and Supplementary Fig. 4a). Cytoplasm-nuclear isolation assays confirmed that RPLP1 predominantly localizes in the cytoplasm in uninfected control cells, less exists in the nucleus. HIV-1 infection caused it translocation to the nucleus in infected cells (Fig. 4b and Supplementary Fig. 4b). Notably, RPLP1 lost its antiviral ability when the cells were treated with Importazole, a nuclear import inhibitor (Fig. 4c, d). Thus, translocation from cytoplasm to nucleus seems required for RPLP1-mediated suppression of HIV-1 LTR transcription. Collectively, these results suggest that HIV-1 clade B infection induces translocation of RPLP1 from the cytoplasm to the nucleus, where RPLP1 occupies the C/EBPβ motif consequently preventing C/EBPβ binding and suppressing viral transcription.

Subsequently, we further investigated whether the translocation upon viral infection was specific for HIV-1 subtype B by infecting HeLa with different subtypes/groups of HIV-1 or other viruses. Surprisingly, in addition to subtype B, subtype C of group M, and groups N, O, and P of HIV-1 also induced translocation of RPLP1 to the nucleus. However, such translocation was not observed during VSV infection (Supplementary Fig. 4c), which replication was not affected by RPLP1 (Supplementary Fig. 4d). These findings indicated that re-localization alone is necessary but not sufficient for the antiviral activity of RPLP1.

## α-helices 2 and 4 of RPLP1 are essential for its HIV-1 inhibitory activity

Since RPLP1 is best known for its importance in protein translation, we investigated whether the antiviral activity of RPLP1 requires this activity. When RPLP2, the partners of RPLP1 in forming ribosomal protein lateral stalk complexes were silenced, the HIV-1 inhibitory effect of RPLP1 was not affected (Supplementary Fig. 5a), suggesting that the antiviral function of RPLP1 is independent of its well-known role as subunit of ribosomal protein lateral stalk complex.

The crystal structure of RPLP1 has been determined to consist four α-helices (Fig. 5a). To define domains critical for HIV-1 inhibition, RPLP1 mutants containing deletions of each of these four α-helices were constructed (Fig. 5b). RPLP1 mutants lacking α-helix 2 (Δα2) or 4 (Δα4) lost the ability to inhibit HIV-1 production, while deletion of α-helix 1 (Δα1) or 3 (Δα3) had no disruptive effect (Fig. 5c, d). Luciferase assays confirmed that α-helices 2 and 4 in RPLP1 are required for inhibition of HIV-1 LTR activity (Fig. 5e, f). DNA binding ELISA assays showed that, RPLP1 mutants lacking α-helices 2 or 4 lost the ability to reduce the levels of C/EBPβ bound to target DNA sequences (Fig. 5g). In agreement with this, ChIP analysis confirmed that LTR-RPLP1 interaction was not affected by deletion of α-helix 1 or 3, while RPLP1 mutants lacking helix 2 or 4 lost their ability to bind the HIV-1 LTR (Fig. 5h). Impairment of C/EBPβ enrichment on HIV-1 LTR induced by RPLP1 was not due to downregulation of C/EBPβ expression (Supplementary Fig. 5b). Taken together, the results showed that α-helices 2 and 4 of RPLP1 are essential for its anti-HIV-1 activity.

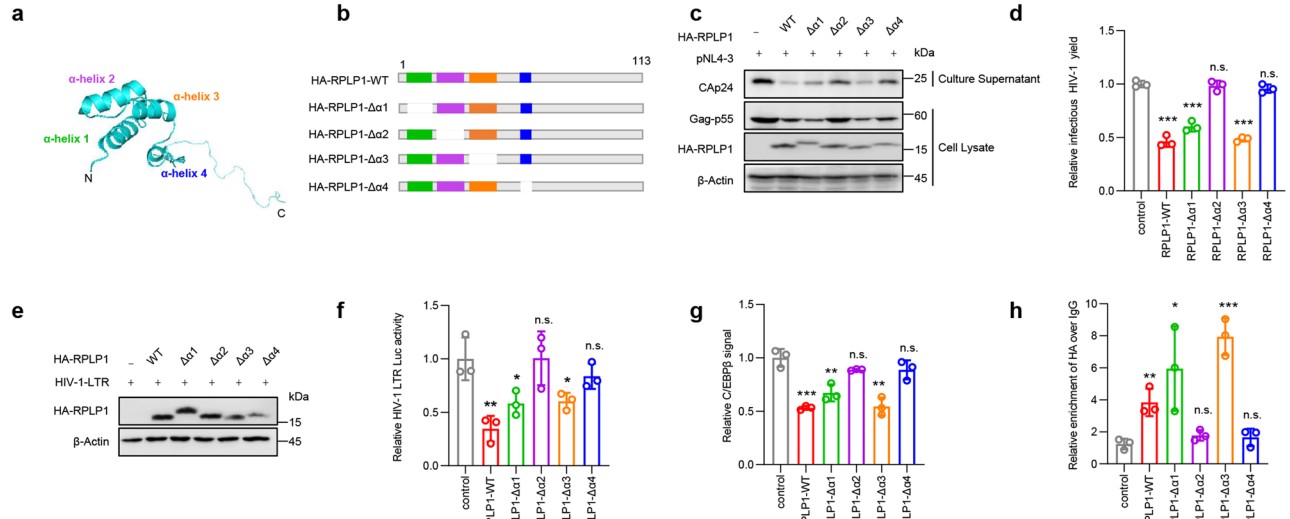

**Fig. 5 | α-helixes 2 and 4 of RPLP1 are essential for its HIV-1 inhibitory activity. a** Crystal structure of RPLP1 published by Lee *et al.* (PDB accession number 3BEH)[57]. The α helixes of RPLP1 were indicated. **b** Sketch map of RPLP1 and its mutants. HA-RPLP1 was mutated to delete each of the four α-helixes. **c**, **d** α-helix 2 or α-helix 4 deletion of RPLP1 (HA-RPLP1-Δα2 or HA-RPLP1-Δα4) lost their inhibition on HIV-1 replication. HEK293T cells were transfected with indicated plasmids for 48 hours. Cells and supernatant were then harvested for immunoblotting (**c**), and infectious virus yield was quantified using the TZM-bl reporter cell infectivity assay (**d**). **e**, **f** HA-RPLP1-Δα2 or HA-RPLP1-Δα4 lost their suppression on HIV-1 LTR activity. HEK293T cells were transfected with indicated plasmid for 48 hours. Cell lysates were then immunoblotted (**e**) and LTR activity was measured (**f**). **g** α-helixes 2 and 4 of RPLP1 were required for the RPLP1 impairment on C/EBPβ-binding.

HEK293T cells were transfected with HA-RPLP1 or its mutants, then harvested after 48 h for isolation of nuclear proteins. Normalized amounts of nuclear extracts were subjected for the TransAM C/EBPβ binding assay (ActiveMotif). **h** RPLP1 mutants lacking α-helix 2 or α-helix 4 lost their ability to bind HIV-1 LTR. HEK293T cells were transfected with HA-RPLP1 or its mutants for 24 hours, then infected with HIV-1 NL4-3 virus for 2 days. ChIP assays were conducted using anti-HA antibody or IgG as a control, followed by RT-qPCR with primers targeting C/EBPβ binding sites. Immunoblots in (**c** and **e**) are representative of three independent experiments. Quantification in (**d**, **f**–**h**) was shown as means ± SDs from three independent experiments. *P* values were calculated by the two-tailed Student's *t* test (**d**, **f**–**h**). *$P < 0.05$, **$P < 0.01$, ***$P < 0.001$, n.s. denotes no significance. Source data are provided as a Source Data file.

## RPLP1 plays a role in maintaining HIV-1 latency

Given that HIV-1 latency is associated with silencing of viral transcription, we explored the role of RPLP1 in HIV-1 latency and reactivation. The cell line C11, which was derived from Jurkat T cells[43], and harbors a latent HIV-1 NL4-3 provirus encoding the green fluorescent protein (GFP) was employed. Higher level of RPLP1 in C11 cells than in Jurkat T cells productively infected with HIV-1 NL4-3 were observed (Fig. 6a). To evaluate the effect of RPLP1 on HIV-1 reactivation from latency, we knocked down the endogenous RPLP1 in C11 (Fig. 6b). After the latent reactivation agent, phorbol 12-myristate 13-acetate (PMA) or sub-eroylanilide hydroxamic acid (SAHA) treatment for 48 h, the cells were harvested to detect the virus reactivation by quantifying the GFP (+) cells, as well as MFI, with flow cytometry. We observed that knockdown of RPLP1 induced the reactivation of latent viruses, and promoted the activating effect of PMA (Fig. 6c, d) and SAHA (Fig. 6e, f) without cytotoxic effects (Supplementary Fig. 5c). Similarly, when silencing endogenous RPLP1 in another latent cell line, ACH-2[44], latent HIV-1 was significantly reactivated as monitored by increased CAp24 expression (Fig. 6g) without cytotoxic effects (Supplementary Fig. 5d). We subsequently investigated the expression of RPLP1 when primary CD4+ T cells were activated. Notably, resting CD4+ T cells obtained from HIV-negative study participants exhibited robust expression levels of RPLP1, which were observed to decline upon stimulation with phytohemagglutinin-M (PHA-M) or anti-CD3/CD28 antibodies to activate CD4+ T cells (Fig. 6h) Notably, we further collected cART-treated study participants living with HIV-1, who had undergone cART for more than 6 months and had undetectable plasma viral loads, and isolated primary resting CD4+ T cells. Upon nucleofected with siRNA against RPLP1 (Supplementary Fig. 5e), latent HIV-1 virus suppressed by cART significantly reactivated (Fig. 6i). These data support that endogenous RPLP1-mediated suppression of HIV-1 transcription promotes viral latency in CD4+ T cells.

## Discussion

In this study, we found the high expression of RPLP1 in LTNPs by mass spectrometry and identified RPLP1 as a transcriptional silencer of HIV-1 group M subtype B strains which require the transcriptional factor C/EBPβ[39], but not of HIV-1 clade C as well as groups N, O, and P strains. Three conserved C/EBP binding sites are located in the HIV-1 subtype B LTR and C/EBP proteins are required for basal and activated levels of LTR transcription in macrophages/monocytes[45,46]. Further investigation revealed that RPLP1 occupies the C/EBPβ binding sites in the LTR of subtype B resulting in HIV-1 transcriptional silence. Meanwhile, we observed a modest impact of RPLP1 on HIV-1 replication and latency in general, potentially attributed to the mild influence of C/EBP sites in CD4+ T cells. Notably, taking the high expression of RPLP1 in LTNPs and the essential role of transcription in HIV-1 latency into account, RPLP1 might play potential role in non-progressive infection and latency of HIV-1.

The C/EBP family which contains six members called C/EBPα, -β, -γ, -δ, -ε, -ζ belongs to a large group of basic region leucine zipper (bZIP) transcription factors[39]. The promoters of various RNA and DNA viruses contain C/EBP binding sites[45–52]. The human HBV genome contains a liver-specific enhancer element (designated enhancer II) that is transactivated by C/EBP[51]. C/EBP proteins also contribute to Epstein-Barr virus (EBV) lytic gene expression and replication[49]. The γ-inducible protein 16 (IFI16) and its family members PYHIN1 and IFIX have been reported to restrict HIV-1 by sequestering the transcription factor Sp1[14,53], showing that interfering with the function of the transcription factors required for HIV-1 replication might allow to control HIV-1 replication. Therefore, the disruption of C/EBP proteins binding to viral LTR or the promotor of key viral proteins might be a promising target to inhibit or control viral infection.

RPLP1 is a subunit of the ribosomal stalk complex. Previous studies have reported that specific ribosomal proteins are implicated as

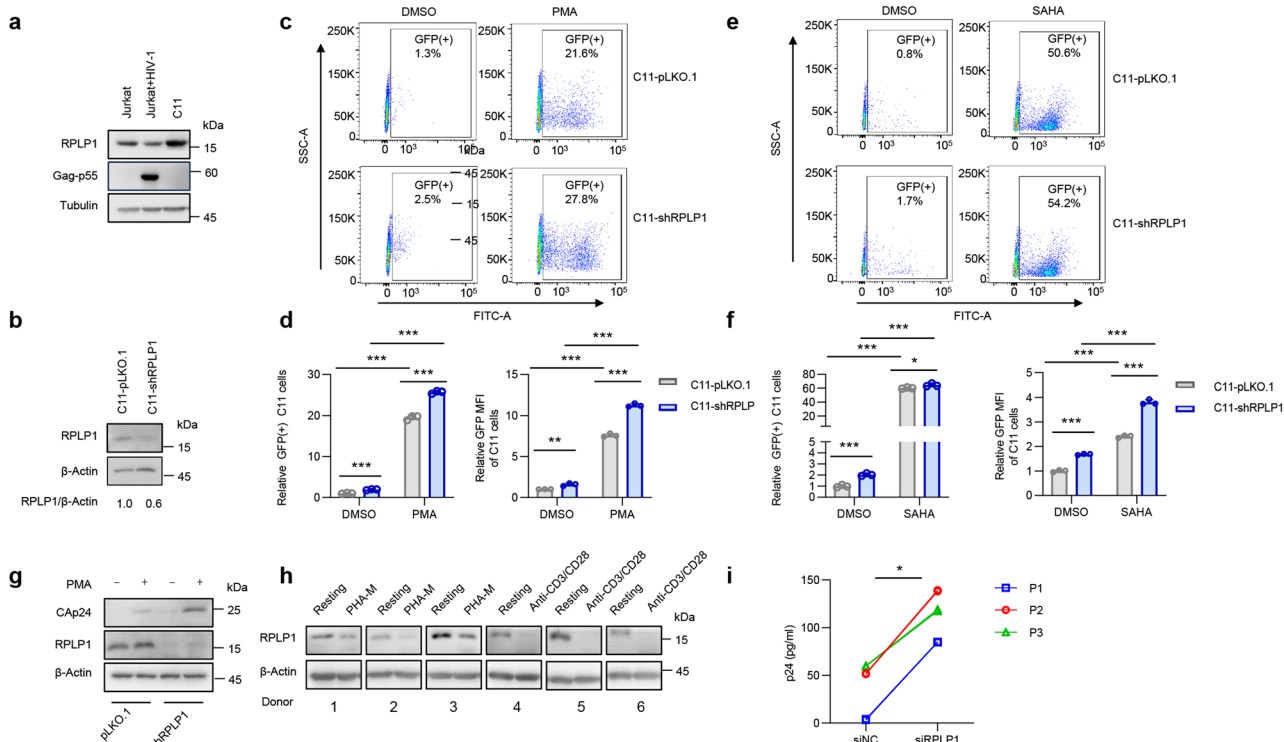

**Fig. 6 | RPLP1 maintains HIV-1 latency. a** High level RPLP1 maintains HIV-1 latency. RPLP1 levels in Jurkat, acutely HIV-1-infected Jurkat (Jurkat+HIV-1) and latently HIV-1-infected Jurkat (C11) cells were detected with immunoblotting. **b–f** Silencing of RPLP1 reactivates latent HIV-1 in C11 cells. Immunoblotting analyses of RPLP1 level in RPLP1-sh or control C11 cells (**b**). RPLP1-sh or control C11 cells were treated with PMA (1 μM) for 48 h, and GFP-positive cells were detected using flow cytometry (**c**). The ratio of GFP-positive cells and MFI relative to untreated control C11 cells in panel (**c**) was calculated (**d**). RPLP1-sh or control C11 cells were treated with SAHA (10 mM) for 48 h, and GFP-positive cells were detected using flow cytometry (**e**). The ratio of GFP-positive cells and MFI relative to untreated control C11 cells in panel (**e**) was calculated (**f**). **g** Silencing of RPLP1 reactivates latent HIV-1 in ACH-2 cells. RPLP1-sh or control ACH-2 cells treated with PMA for 48 h, RPLP1 and HIV-1 CAp24 were measured by immunoblotting. **h** RPLP1 levels were reduced upon CD4+ T cells activation. CD4+ T cells isolated from HIV-negative participants (*n* = 6

donors) were activated with PHA-M (5 ng/ml) or anti-CD3/CD28 antibody-coated microbeads, then harvested to detect RPLP1 levels. **i** Silencing of RPLP1 increases HIV-1 reactivation in primary CD4+ T cells. CD4+ T cells isolated from cART-treated study participants living with HIV-1 (*n* = 3 donors) were nucleofected with siRNA against RPLP1, HIV-1 reactivation in CD4+ T cells treated with PHA-M (5 ng/ml) was detected by measuring CAp24 levels in supernatants with ELISA. P1-P3 demoted three participants. Immunoblots in (**a**, **b**, **g**, **h**) are representative of three independent experiments. Quantification in (**d**, **f**, **i**) was shown as means ± SDs from three independent experiments. The densitometric analysis of protein levels in (**b**) is shown as the mean value (*n* = 2) relative to β-Actin. *P* values were calculated by the two-tailed Student's *t* test (**d**, **f**, **i**). *$P < 0.05$; ** $P < 0.01$; *** $P < 0.001$, n.s. denotes no significance. See also Supplementary Fig. 5. Source data are provided as a Source Data file.

translational regulators of viruses. Examples include RPS25, that is required for the internal ribosome entry site (IRES) functions of the hepatitis C virus (HCV) and the cricket paralysis virus (CrPV) in a cap-independent translation manner[54]. The large ribosomal subunit protein rPL40 is also required for cap-dependent translation of vesicular stomatitis virus (VSV), measles virus and rabies virus[55]. Recent studies reported that RPLP1/2 is essential for the replication of several mosquito-borne flaviviruses including dengue (DENV), Zika and yellow fever viruses (YFV)[25,26]. Here, we demonstrate the antiviral function of RPLP1 as a transcriptional silencer of HIV-1 group M subtype B strains but not of HIV-1 clade C strains that currently dominate the AIDS pandemic[56]. It is worth further investigating whether RPLP1 exerts broad antiviral effects.

RPLP1 mainly localizes in the cytoplasm, while the expression of the HIV-1 LTR occurs in the nucleus. Confocal microscopy and cytoplasm-nuclear isolation assays both demonstrate that HIV-1 subtype B infection induces nuclear translocation of RPLP1. Whether nucleic RPLP1 can inhibit the virus replication depends on the viral transcription requirement for C/EBPβ. Meanwhile, based on our observation of the differential re-localization of RPLP1 between HIV-1 and VSV infection, we propose that the translocation of RPLP1 to the nucleus may be associated with viral life cycle. The HIV-1 life cycle necessitates nuclear entry, during which RPLP1 may be accompanied

into the nucleus; in contrast, VSV infection does not involve nuclear entry and therefore does not result in RPLP1 translocation to the nucleus. Importantly, RPLP1 loses its inhibitory effect on HIV-1 when treated with the nuclear import inhibitor Importazole. Thus, RPLP1 translocation into the nucleus upon HIV-1 infection is essential for its inhibitory effect. The underlying mechanism of HIV-1-induced translocation of RPLP1 warrants further investigation.

In addition, we found that α-helixes 1 and 3 domains of PRLP1 are not required for HIV-1 LTR binding and inhibition (Fig. 5), the latter has been reported to play a vital role in anchoring the RPLP1/RPLP2 heterodimer to RPLP0[57]. Along with the finding that silencing of RPLP2 did not affect the anti-HIV effect of RPLP1 (Supplementary Fig. 5), thus, HIV-1 inhibition by PRLP1 is not dependent on P complex assembly. Intriguingly, we found that α-helixes 2 and 4 domains of RPLP1 were responsible for LTR binding (Fig. 5).

We note the technical limitations of the analysis in this study. First, the uneven replications of study participants, due to sample rarity, should be taken into consideration. Therefore, based on the valuable clues provided by mass data, we performed some biological experiments to verify the important role of RPLP1, the most up-regulated host factor in LTNPs, in HIV-1 replication. Secondly, due to individual differences, further investigations into other factors may benefit us in deeply exploring the mechanism of HIV-1 latency.

In summary, HIV-1 infection causes the translocation of RPLP1 from cytoplasm to the nucleus, where occupation of C/EBPβ binding sites by RPLP1 impedes the interaction of transcription factor C/EBPβ with LTR, thereby inhibiting the HIV-1 transcription (Supplementary Fig. 6). As a transcriptional inhibitor, whether RPLP1 is antagonized by some viral proteins and whether RPLP1 inhibits other viruses required for C/EBPβ protein for transcription replication is worth to be further investigated.

## Methods

### Study participants living with HIV-1 and HIV-negative study participants

A cohort of 10 LTNP study participants living with HIV-1 from Beijing Youan Hospital were referred to us: HIV-positive, ART-naive, normal and stable CD4 cell counts for more than 10 years. Among them, to select the most representative LTNPs, the inclusion criteria that HIV-1 RNA levels of less than 50 copies/mL of plasma for more than 15 years without ART was adopted[58]. Meanwhile, the ones who had viral infection other than HIV-1 were also excluded. Finally, three LTNPs meet the inclusion criteria. The study participants ranged in age from 55 to 65 years. Among them, two participants were infected by IV injection, and one participant was infected by heterosexual contact. Two demographic characteristics-matched RPs were enrolled, and their demographic and laboratory characteristics information was listed in Supplementary Table 1. Besides, three HIV-negative study participants and three cART treated study participants living with HIV-1 were also enrolled to isolate CD4+ T cells to investigate the impact of RPLP1 on HIV-1 replication and latency, and the characteristics of cART-treated study participants living with HIV-1 was listed in Supplementary Table 2.

### Proteomic analysis

The whole blood samples were obtained from 3 LTNPs and 2 RPs study participants living with HIV-1, and the CD4+ T cells were isolated using anti-CD4-specific antibody-coated microbeads (130-045-101, Miltenyi Biotec, Germany) according to the manufacturer's instructions. Protein from each sample was reduced and alkylated with 100 mM DTT for 30 min at 56 °C and 50 mM iodoacetamide (IAM) for 30 min at room temperature, respectively. Protein digestion was processed with trypsin (1:30) (Promega) at 37 °C overnight. Then, the peptides derived from RP samples were labeled with TMT tags as 126, 127, and LTNP samples with TMT tag as 128, 129, 130 as per the manufacturer's instruction (Catalog # 90110, Thermo Fisher Scientific). The labeling reaction was terminated by adding 5% hydroxylamine followed by sample pooling. The labeling peptides were fractionated into 24 fractions using high pH reversed-phase HPLC. These fractions were analyzed by synchronous precursor selection (SPS)-MS3 method on an Orbitrap Fusion Lumos Tribrid mass spectrometer (Thermo Scientific, Bremen, Germany). Peptides were resolved on an analytical column (150 μm × 150 mm) using a linear gradient of 7% to 25% of solvent B in 38 min, 25-38% in next 20 min. Next, the gradient was quickly ramped to 95% in 4 min and stayed there for 10 min. All the experiment was performed with three technical replicates.

The data were acquired in a data-dependent mode. The precursor MS scans (from $m/z$ 400–1500) and MS3 scans (from $m/z$ 100–500) were acquired in the Orbitrap at a resolution of 120,000 and 60,000. The MS2 scans were acquired in the ion trap. The isolation window was set to 0.7 and 2 for MS1 and MS2, respectively. Automatic Gain Control (AGC) target was $4 \times 10^5$ for MS1, $1 \times 10^4$ for MS2, and $5 \times 10^4$ for MS3, respectively. Dynamic exclusion was set with exclusion duration of 15 s.

For protein identification, the raw data were processed with the software Proteome Discover (version 2.1.1.21, Thermo Scientific) against UniProt human protein database (released on Dec 2016). Trypsin was designated as the protease, with a maximum allowance of two missed cleavages. The search parameters included carbamidomethylation at cysteine residues, TMT 6-plex (+229.163) modification at N-termini of peptides and lysine residues as fixed modification while oxidation of methionine was set as variable modification. MS and MS/MS mass tolerances were set to 10 ppm and 0.5 Da, respectively. For protein quantitation, top3 unique and razor peptides with reporter ion mass tolerance of less than 20ppm were used. Peptide precursor ion isolation purity should be > 75% and the summed S/N of all channels R 200. The false-discovery rates (FDR) were controlled at <1%. Normalization was performed against the total peptide amount. The mass spectrometry data generated in this study have been deposited in the ProteomeXchange Consortium under accession code PXD050294 (https://www.iprox.cn//page/project.html?id=IPX0008310000).

### Identification of differentially expressed proteins

Proteome Discoverer performed the global equal sum normalization with protein scaling, which scales the normalized summaries across each protein to generate the protein ratios with an average of 100. This normalized data encompassed a total of 4893 proteins. Proteins with missing value in more than 1 runs were excluded from the dataset. This filtering process resulted in a reduced set of 3616 unique proteins. The normalized data were then log2 transformed. Since this data had multiple MS runs and missing values, we used limma[59–61] package (version 3.58.1) to identify proteins whose mean levels are significantly different between LTNPs and RPs. We first used lmFit function in limma package to fit an additive linear model with a fixed group effect and a fixed MS run effect. We then used the Empirical Bayes procedure in limma to combine the estimates of random variation across all the proteins in a moderated $t$-statistic. In brief, we extracted and combined the abundances from three runs. Subsequently, for each run, if a protein has a value of NA in any of the five samples, we set the expression value of this protein to NA in all five samples. Next, each protein level between samples was normalized to an average of 100. Following this, proteins with missing data in more than one run were filtered out, and differentially expressed proteins were identified using the limma package (version 3.58.1), based on transformed log2 values with group and runs as covariates. Then, we mapped UniprotIDs to gene symbols on the Uniprot website. Finally, a volcano plot was generated using ggVolcano package based on logFC and $P$ Value. The detailed descriptions along with the R code used in this study has been publicly available online (https://github.com/wuruihongJilin/RPLP1study.git). The differentially expressed proteins were defined as those with a fold change greater than 2, $P$ value less than 0.05.

### Plasmid construction

The full-length RPLP1 was amplified by PCR with total cDNA of human peripheral blood mononuclear cells (PBMCs) as the template and cloned into VR1012 vector between the *Eco*RI and *Bam*HI restriction sites with C-terminal Flag tag. HA-RPLP1 was constructed using Flag-RPLP1 as template to mutate Flag tag with hemagglutinin (HA) Tag. HA-RPLP1deletion mutants (Δα1, Δα2, Δα3, Δα4) were generated from HA-RPLP1 by PCR-based site-directed mutagenesis. The expression vector of C/EBPβ with Myc tag was synthesized by Shanghai Generay Biotech Company, China. The plasmids pNL4-3, pNL4-3-ΔEnv-GFP were obtained from the AIDS Research and Reference Reagents Program, Division of AIDS, National Institute of Allergy and Infectious Diseases (NIAID), National Institutes of Health (NIH). Tat- HA was generated by amplifying two exons of Tat using pNL4-3 as the template, incorporating a HA epitope tag at its 3′ terminus, then the resulting fragments were ligated and cloned into the VR1012 vector. HIV-1-LTR-luciferase was constructed by amplification of the LTR using pNL4-3 as the template, and then cloned into the pGL3-basic vector (Promega, Madison, WI)[62]. Various infectious molecular clones (IMCs) of proviral HIV- were constructed by Beatrice Hahn lab (University of Pennsylvania, USA.)[40–42]. LTR transcription factor binding sites mutants (ΔNF-κB,

mut-NF-κB, mut- SpI, mut-C/EBPβ, mut-TCF-1α, mut-USF) were generated from pHIV-1-LTR-luciferase by PCR-based site-directed mutagenesis. The LTR-luciferase vector of HIV-1 CH167 was synthesized by Shanghai Generay Biotech Company, China, and the chimeric LTR-luciferase vector of CH167 containing C/EBPβ binding sites were generated by PCR-based site-directed mutagenesis.

All the primers used for plasmid construction are listed in Supplementary Table 3.

## RNA extraction and RT-qPCR
RNA was isolated using the Trizol Reagent (15596-026, Thermo, Waltham, MA, USA) according to the manufacturer's protocol, then RNA was treated with RQ1 RNase-free DNase (M6101, Promega) to remove DNA. cDNA synthesis was performed using a Transcriptor First Strand cDNA Synthesis kit (MR05101M, Monad, Wuhan, China). A total of 250 to 1000 ng of total RNA was used as a template for each cDNA synthesis reaction, and samples containing only $H_2O$ was considered as blank samples. RT-qPCR was executed on the Roche 480 instrument with the MonAmp ChemoHS qPCR Mix (MQ00401S, Monad). The primer sequences of RT-qPCR are listed in Supplementary Table 4.

## Cells and antibodies
Human embryonic kidney 293T (HEK293T) (catalog no. CRL-11268), HeLa (catalog no. CCL-2) and TZM-bl (catalog no. PTA-5659) cells were obtained from American Type Culture Collection (ATCC; Manassas, VA, USA) and cultured in Dulbecco's modified Eagle's medium (11995065, Thermo) supplemented with 10% fetal bovine serum (FBS; ST30-3302, PAN Seratech, Aidenbach, Germany). The HIV-1 latent C11 cell line was a gift from H. Z. Zhu (The College of Life Science, Fudan University). ACH-2 (catalog no. 349) and MT-4 (catalog no. 120) cells were obtained from the AIDS Research and Reference Reagents Program, Division of AIDS, NIAID, NIH. Jurkat (ATCC catalog no. TIB-152), H9 (ATCC catalog no. HTB-176), THP1 (ATCC catalog no. TIB-202), C11 and MT-4 cells were cultured in RPMI 1640 medium supplemented with 10% FBS and Penicillin-streptomycin Solution (03-031-1B, Biological Industries, Israel). ACH-2 cells were maintained in RPMI 1640 medium supplemented with 10% heat-inactivated FBS, HEPES (15630-080, Gibco, Grand Island, NY, USA). The PBMCs were isolated through Ficoll density gradient centrifugation, and the CD4 + T lymphocytes were then purified from the PBMCs with anti-CD4-specific antibody-coated microbeads (130−045−101, Miltenyi Biotec, Teterow, Germany) according to the manufacturer's protocol. All cell lines were maintained at 37 °C in a humidified atmosphere containing 5% $CO_2$.

The antibodies used in this study are listed as follows: Rabbit polyclonal anti-HA(SG77) (#715500, Thermo), Mouse monoclonal anti-Myc (clone 4A6) (#05-724, Millipore, Billerica, MA, USA), Mouse monoclonal anti-Flag (M2) (#F1804, Sigma, St. Louis, MO, USA), Mouse monoclonal anti-β-Actin (#A00702, GenScript Corporation, PISCATAWAY, NJ, USA), Rabbit polyclonal anti-Histone H3 (#A01502, GenScript Corporation), Mouse monoclonal anti-p24 (catalog no. 1513; AIDS Research and Reference Reagents Program [ARRRP], USA), Mouse monoclonal anti-β-Tubulin (MG7) (#RM2003, Ray Antibody Biotech, Beijing, China), Mouse monoclonal anti-GAPDH(MC4) (#RM2002, Ray Antibody Biotech), Rabbit polyclonal anti-RPLP1 (#21636-1-AP, Proteintech, Rosemont, IL, USA), Rabbit polyclonal anti-RPLP2 (ab154958, Abcam, Cambridge, UK), Rabbit polyclonal anti-C/EBPβ (D155298-0025, BBI, China), Peroxidase AffiniPure Goat Anti-Mouse IgG (H + L), (#115-035-062, Jackson, West Grove, PA, USA), Peroxidase AffiniPure Goat Anti-Rabbit IgG (H + L), (#111-035-045, Jackson).

## Transfection and immunoblotting analysis
HEK293T cell was seeded in 12-well plates at a monolayer density of 2.5 × 10⁵ cells/well and cultured overnight, and then transfected with indicated plasmids using Lipofectamine 2000 Reagent (# 11668019,

Thermo) according to the manufacturer's protocol. Transfection of siRNAs were performed using Lipofectamine RNAiMAX Reagent (# 13778150, Thermo). Human CD4+ T cells isolated from peripheral blood were nucleofected using an Amaxa human T-cell Nucleofector kit (VPA1002, Lonza, Basel, Switzerland) with the program U-014.

For immunoblotting analysis, cells were harvested and lysed with RIPA buffer containing 1 M Tris-7.8, 1 M NaCl, NP40 and 0.5 M EDTA. The samples were heated at 100°C, followed by separation on 12% SDS-PAGE gels. Proteins transferred onto PVDF membranes were then incubated with antibodies against indicated proteins and visualized using the Meilunbio fgsuper sensitive ECL luminescence kit (# MA0186-1, Meilunbio, Meilun Biotechnology co. Ltd, Dalian, China).

## Virus infection and reactivation
The viruses were produced by transfecting HEK293T cells with the wild-type HIV-1 plasmid (pNL4-3) or pNL4-3-ΔEnv-GFP and VSVG. The supernatants were collected at 48 h post-transfection by centrifugation. For virus infection, cells were incubated with the supernatants as mentioned above for 4-6 h, then washed with PBS and cultured with fresh medium. The cells were collected after 48 h for the subsequent experiments.

For virus reactivation assay, HIV-1 latently infected cells ACH-2 and C11, which stably knocking down endogenous RPLP1 expression, were treated with 1 µM phorbol 12-myristate 13-acetate (PMA) or 10 mM suberoylanilide hydroxamic acid (SAHA). The C11 cells lines were measured using flow cytometry (FACSCalibur; BD, Franklin Lakes, NJ, USA) to detect the GFP-positive cells with green fluorescence (FL1, 488 nm), and the activation of latent HIV-1 in ACH-2 cells were determined by immunoblotting to detect p24 levels.

## Stable cell lines
To generate stable RPLP1-overexpressing cells, HA-tagged RPLP1 was cloned into pLVX-IRES-neo (Clontech Laboratories Inc., San Francisco, CA, USA). Similarly, short hairpin RNAs against RPLP1 were subcloned into pLKO.1-puro vector to knock down endogenous RPLP1 expression. Using a four plasmids transient-cotransfection method, the lentivirus was produced by transfecting HEK293T cells with Lipofectamine 2000 (Invitrogen) and subjected to infect target cells for 2 days. Then the cells were selected with 1 µg/ml puromycin (# S7417 Selleckchem, Houston, TX, USA) and confirmed with immunoblotting.

## Luciferase assay
HEK293T cells transfected with LTR or TZM-bl cells infected with viruses were collected at 48 h post transfection or infection, and lysed with lysis buffer at room temperature for more than 30 min. Luciferase assay was performed using the Dual-luciferase reporter assay system (# E1910, Progema, Madison, Wisconsin, USA) according to the protocol with a GloMax 20/20 luminometer (Promega).

## Nuclear-cytoplasm separation
HIV-1 infected or uninfected Jurkat, H9, MT-4 or HIV-negative study participants derived CD4+ T cells were collected and lysed with hypotonic lysis buffer at room temperature for 1 h. Then 1 µl of digitonin was added to decomposed the nuclear membrane. Nuclear were separated by centrifugation, and supernatants were cytoplasmic. The proteins expression was detected by immunoblotting with indicated antibodies.

## Flow cytometry
Flow cytometry was performed to detected the antiviral effect of cells which overexpressed or knockdown RPLP1 gene infected with VSV-G pseudotyped HIV-1 NL4-3-GFP constructs, and the percentage of GFP positive cells in cells population were determined after infected 16 h. Flow cytometry was also used to detected the level of C11 cells latency reactivation that stimulate with PMA or SAHA.

## Confocal microscopy

The subcellular localization of RPLP1 were detected by confocal microscopy. HeLa cells were infected by NL4-3-EGFP virus and changed with fresh cultured medium after 4-6 h. Forty-eight hours later, cells were treated with 4% paraformaldehyde solution at room temperature for 10 mins and washed 3 times with PBS, followed by treated with 0.4% Triton-X-100 for 10-20 mins. After being washed with PBS 3 times, 1% BSA was added to block the nonspecific binding for 30 min. RPLP1 were stained using RPLP1 polyclonal antibody while nuclear was stained using DAPI. The confocal microscopy for CD4+ T cells was similar as HeLa cells, except that CD4+ T cells needs to be immobilized in PolyPrep slides (Sigma-Aldrich) for 15 min before fixation with paraformaldehyde. Images were obtained with Olympus FV 3000 confocal imaging system, and RPLP1 localized in the nucleus was quantified by measuring the nucleic red signal intensity using ImageJ software.

## C/EBPβ DNA binding assay

HEK293T cells transfected with constructs of HA-RPLP1 or its mutants were collected to isolate the nuclear proteins at 48 h post-transfection according to the protocol. In brief, cells were washed with cold PBS and harvested in 1 ml cold phosphatase inhibitor buffer. Then the mixture was centrifuged at $300 \times g$, 5 min at 4 °C and the cell pellet was resuspended in 1 ml hypotonic buffer, followed by incubating on ice for 15 min. After addition of 50 μL 10% NP40, the mixture was centrifuged at $20,000 \times g$ for 1 min at 4 °C to remove the supernatant. Subsequently, the pellet was resuspended with 50 μl Complete Lysis Buffer AM1 (#44196, ActiveMotif, Carlsbad, CA) and incubated for 30 min at 4 °C. The homogenate was centrifuged at $20,000 \times g$ for 10 min at 4 °C, and the supernatant was nuclear extract. BCA Protein Assay Kit (#P00125, Beyotime, Changchun, China) was used to determine protein concentrations and 20 μg of the nuclear extract was used to examine C/EBPβ binding to target DNA sequences following the experiment and specific steps refer to the Transcription Factor Assay Kit (#44196, ActiveMotif, Carlsbad, CA, USA).

## Chromatin immunoprecipitation (ChIP)

ChIP analysis was performed with a chromatin immunoprecipitation kit (17-371, Millipore) according to the manufacturer's protocol. Briefly, cells infected with HIV-1 were cross-linked with 1% formaldehyde (Sigma-Aldrich) for 10 min at room temperature and then quenched by 125 mM glycine for 5 min. The cell pellets were then collected by centrifugation at $700 \times g$ for 5 min at 4 °C, followed by washing with cold PBS three times and resuspending in 1 ml SDS lysis buffer containing 1× protease inhibitor cocktail II. The cell lysates were sheared on wet ice using a Cole-Parmer instrument with 5 sets of 10-s pulses to obtain the DNA fragment of 150 to 900 bp in length and were centrifuged at $12,000 \times g$ for 10 min at 4 °C. Nuclear extracts were incubated with the indicated antibodies overnight at 4 °C with rotation, and subsequently incubated with 60 μl of protein G agarose for another 2 h at 4 °C with rotation. The immunoprecipitated DNA was detected by RT-qPCR, and the primers are listed in Supplementary Table 4.

## Chemical synthesis of siRNA

To knock down indicated genes, short interfering RNA (siRNA) of RPLP1, RPLP2, and C/EBPβ were synthesized by RiboBio (Guangzhou, China).

## Latent HIV-1 reactivation of cART-treated study participants living with HIV-1

$1.0 \times 10^6$ CD4+ T cells isolated from cART-treated study participants living with HIV-1 were nucleofected with control or siRNA-targeted RPLP1, and cultured for another 2 days. Then the cells were collected for immunoblotting and stimulated with phytohemagglutinin M (PHA-M) (5 ng/ml; Sigma-Aldrich) for 7 days. The reactivation of HIV-1 was detected by measuring the p24 amount in the supernatants with ELISA.

## Ethics statement

Collection of blood samples from hospitalized study participants was approved by the ethics committee of the First Hospital of Jilin University (license number 22K086-001). Written informed consent was obtained from study participants.

## Reporting summary

Further information on research design is available in the Nature Portfolio Reporting Summary linked to this article.

## Data availability

The mass spectrometry data generated in this study have been deposited in the ProteomeXchange Consortium under accession code PXD050294 (https://www.iprox.cn//page/project.html?id=IPX0008310000). All other data are available in the main text or in the supplemental information. Source data are provided with this paper.

## Code availability

R code used in this study has been publicly available online (https://github.com/wuruihongJilin/RPLP1study.git).

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

## Acknowledgements
The authors thank C.Y. Dai for providing critical reagents. We thank F. Bibollet-Ruche and B.H. Hahn for providing HIV-1 IMCs. We thank H. Z. Zhu for C11 cell line. We thank Beijing Youan Hospital for helping us collect LTNP samples. We thank Dr. Ruihong Wu in the core facility, and Bioinformatics Laboratory of The First Hospital of Jilin University for the training and generous sharing of experiences and codes. This work was supported by funding from the National Natural Science Foundation of China (81930062) and the National Key R&D Program of China (2021YFC2301900 and 2021YFC2301904) to W.Z., the Science and Technology Department of Jilin Province (20210101300JC) to C.H., the German Research Foundation (CRC 1279) and an ERC Advanced grant (Project 101054456) to F.K.

## Author contributions
W.Y. conducted experiments. H.W. and Z.L. provided technical and material support. L.Z. and J.L. performed the mass spectrometry analysis. W.Z. and C.H. analyzed and discussed the data and wrote the manuscript. F.K. provided reagents and insightful advice and edited the manuscript. All authors have read and approved the final manuscript.

## Competing interests
The authors declare no competing interests.
