## [Peer Review File · Nature Communications]

REVIEWER COMMENTS

Reviewer #1 (Remarks to the Author):

Authors Yang et al in their manuscript “RPLP1 restricts HIV-1 transcription by disrupting C/EBP β binding to the LTR ” present a functional study of a particular protein subunit of the ribosome, the ribosomal protein lateral stalk subunit P1 (RPLP1) and show how this protein inhibits HIV-1 replication. The authors employ a number of biochemical and molecular biology techniques to support their claim, including mass spectrometry, reporter cell lines, ChIP-seq, and flow cytometry. In this review, I focus in particular regarding proteomics, specifically the TMT-based mass spectrometry. Based on their methods and figures, there is not enough detail provided to determine whether the authors have performed suitable acquisition of their mass spectrometry proteomics data and have analyzed their data appropriately.

Major concerns

- Please add additional proteomics methods information, e.g. reduction, alkylation, digestion, LC-MS method, and analysis details so that others can reproduce the results. Additionally, please add data repository details to the main text as it is currently only found in the reporting information, “Mass spectrometry data are available in iProX (IPX0007276000)”, and please make this repository public, as it is not possible currently to assess the mass spectrometry results. If not already in iProX, please also share the SEQUEST HT output files with each TMT channel’s quantified proteins, for reproducing Fig 1a volcano plot.

Minor concerns

- RPLP1 is a very small protein, only 114 amino acid residues. Assuming the authors used trypsin to digest their samples, this would produce potentially 3 tryptic peptides, depending on how many missed cleavages were used in the database search. Are the mass spectrometry results using protein parsimony algorithms or razor peptides to determine proteins?

- Could the authors speak to the specificity of their anti-RPLP1 antibody? Perhaps a supplemental figure showing the full Western staining of the cytosol and/or nuclear protein fractions stained with the anti-RPLP1 antibody to show the specificity of binding.

- Fig 3a states that the F2/R2 binding is not significant between IgG and anti-RPLP1, but the figure does look significant albeit by eye. Could the authors provide the raw data values used to generate this figure?

Reviewer #2 (Remarks to the Author):

The manuscript by Yang et al identify RPLP1 as a potential factor that regulates HIV-1 LTR through a mechanism that interferes with C/EBP β binding. Their data suggests that RPLP1 only inhibits clade B viruses, which have conserved C/EBP sites but do not inhibit transcription of other clades. They perform experiments characterizing the potential role of RPLP1 as a transcriptional repressor for the LTR utilizing cell lines, some primary cells and gain and loss of function experiments. In general, most of the effects are modest being approximately two fold. There are limited experiments that address how RPLP1 activity is regulated in CD4+ T cells. The potentially most novel aspect of the paper is the suggestion that RPLP1 limits HIV-1 transcription in long term non-progressors, however, there are no follow-up experiments that address whether RPLP1 limits transcription in cells from this cohort or influences the susceptibility of these cells to different clades providing little insight if this factor is directly contributing to controller phenotype. Specific comments are below.

1. It would have been interesting to use CD4+ cells from their LTNP to determine if cells were more or less resistant to clade B vs other clades. Similarly, they could potentially knock-down RPLP1 in these cells and see if this altered their ability to support HIV-1 infection and transcription. Such experiments would make a stronger case for a role in limiting expression in the context of the non-progressors.
2. The description of the Mass Spec and representation of the volcano plot was confusing. I am assuming this represents samples combined from the three controllers and the two individuals on ART. Also, beyond the Mass Spec, was the overexpression of RPLP1 confirmed? Furthermore, it would have been informative to have more details regarding how RPLP1 expression is regulated in resting and activated CD4+ T cells.
3. There was limited information concerning other proteins that were identified as being overexpressed or differentially reduced in the controllers. Data for the mass spec are not discussed and do not seem to be included in the manuscript. The only protein identified in the volcano plot is RPLP1.
4. The impact of RPLP1, in general, is modest in all their experiments, being about a two-fold change, regardless of knock-down or over-expression. Might this reflect the reported minimal impact of C/EBP sites in CD4+ T cells? This should be discussed.

5. Use of RPLP1 mutations to appreciate the structure and function of this protein in the context of HIV transcription was informative.

6. Minor comment, for experiments in Fig. 1, the MFIs for the flow cytometry should also be reported, especially since their data supports, minimal impact on infection and the largest impact on transcription.

7. Minor comment: They indicate as “data not shown” for cell viability. These data should be included in the manuscript.

Reviewer #3 (Remarks to the Author):

In the manuscript by Zhang and colleagues, the authors describe a surprising role for a ribosomal protein, RPLP1, in HIV-1 infection. The first observation of a role for RPLP1 comes from a proteomics screen comparing CD4 cells from 3 non-progressors and 2 viremic progressors. The authors then describe how the protein is and proviral transcription by binding to the CEBPB binding site at the HIV promoter and thereby competing out the CEBPB transcription factor. The authors lastly test the effect of depleting RPLP1 in 3 study participants living with HIV, and find depletion induces expression of HIV-1.

The results are novel and original. The biochemical work, such as the protein domain deletions and mutation of TF binding sites, is strong and well-founded. However, some of the conclusions and claims need further data and analysis but when substantiated, the study is highly interesting for a general audience. The current manuscript is somewhat limited, as the manuscript is framed in a clinical context, although it mainly relies on results from cell lines. The authors describe several surprising findings that require further in depth studies to be confirmed.

Major comments:

1. Although the initial observation of RPLP1 was done in a small study of CD4 samples of progressors (n=2) and non-progressors (n=3), the strong link that the authors make in the abstract and introduction requires further support. To “show that the ribosomal protein lateral stalk subunit P1 (RPLP1) is expressed at higher levels in LTNP compared to regular progressors (RPs)” (abstract) the authors should test this in a larger independent cohort.

Also, as both of these contrasting groups (LP vs PR) are extremes, what are the levels of RPLP1 in virally suppressed PLWH, representing the majority of cases?

2. Some work is based on primary CD4 cells (1d and 6h-i), but the overwhelming work is done in cancer cell lines, with reporter HIV. Although the effect they observe is expected to have its strongest effect in monocytes/macrophages, as reinforced in the discussion section, this is never tested in primary material. Some of the unexpected key findings, including the RPLP1 translocation to the nucleus should be tested in primary cells and monocytic cells as well.

3. Several unexpected observations are made and they are jointly linked to the subtype B of the HIV-1. Is the nuclear translocation also only observed in this subtype? As this protein redistribution is key to the mechanism, I would like a more thorough characterization. Does nuclear RPLP1 also occur after stresses, other viruses, or other subtypes of HIV?

4. The data underlying Fig 6 h-i, where the authors use primary material from PLWH. Here the data does not match up. They state that 1 million CD4 cells were used, but manage to split the cells for two nucleofections, do a immunoblot but also stimulate and grow cells for another 5 days, which result in massive cell death under normal conditions. In the figure, the observed variation in the p24 levels is close to zero. For reference the variation for comparable method in fig 1d has reasonable variation between measurements. The extremely low levels here also make important to denote the limit of detection. The expected reduction in viability after RPLP1 depletion could lead to this observation.

Given that a minute fraction of CD4 cells in PLWH have intact provirus, these results are highly puzzling, even though they confirm the authors hypothesis.

Also, the HIV subtype of the study participants is not explicitly mentioned, despite its importance. To validate this crucial section, the authors could add study participants with with subtype C to the data underlying fig 6 h-i.

Minor comments:

1. Please use people first language, e.g. "Study participants living with HIV-1" instead of "HIV-1-infected individuals" or "HIV patients",

2. How are the cells affected by depletion of a major component of a ribosomal unit. What is the viability of after knock-down? In Fig 1b, there is an almost complete depletion of RPLP1. When over-expressing, does it affect the cellular phenotype? This protein is already highly expressed.

3. Figure 1d. Why the axes so different between empty vectors?

4. Figure 6g, the RPLP1 effect here seems to be derived from a change in beta active levels, not RPLP1. To make quantitative measurements on an immunoblot with these small differences, an alternative method should be used, such as stain-free gels.

5. The introduction and results section would benefit from being more focused and aligned to the results section.

We would like to thank the reviewers for the valuable feedback regarding our manuscript. Their thorough and insightful comments have allowed us to improve the manuscript significantly. In response to their constructive suggestions, we have conducted additional experiments, implemented necessary modifications to the text and figures, and expanded discussions to offer more profound insights into our findings. Here, we present a comprehensive point-by-point response to address each of the reviewers' comments (underlined):

Reviewers' comments:

Reviewer #1 (Remarks to the Author):

Authors Yang et al in their manuscript “RPLP1 restricts HIV-1 transcription by disrupting C/EBP β binding to the LTR ” present a functional study of a particular protein subunit of the ribosome, the ribosomal protein lateral stalk subunit P1 (RPLP1) and show how this protein inhibits HIV-1 replication. The authors employ a number of biochemical and molecular biology techniques to support their claim, including mass spectrometry, reporter cell lines, ChIP-seq, and flow cytometry. In this review, I focus in particular regarding proteomics, specifically the TMT-based mass spectrometry. Based on their methods and figures, there is not enough detail provided to determine whether the authors have performed suitable acquisition of their mass spectrometry proteomics data and have analyzed their data appropriately.

Response: Thank you so much for your insightful comments, we have added the details regarding mass spectrometry proteomics in material and method (Lines 362-398). This part of work, we cooperated with Lihua Zhang team (CAS Key Laboratory of Separation Science for Analytical Chemistry, National Chromatographic Research and Analysis Center, State Key Laboratory of Medical Proteomics, Dalian Institute of Chemical Physics, Chinese Academy of Science), which is professional in the field of mass spectrometry. We think that these revisions have effectively addressed your concerns.

Major concerns

- Please add additional proteomics methods information, e.g. reduction, alkylation, digestion, LC-MS method, and analysis details so that others can reproduce the results. Additionally, please add data repository details to the main text as it is currently only found in the reporting information, “Mass spectrometry data are available in iProX (IPX0007276000)”, and please make this repository public, as it is not possible currently to assess the mass spectrometry results. If not already in iProX, please also share the SEQUEST HT output files with each TMT channel’s quantified proteins, for reproducing Fig 1a volcano plot.

Response: We thank the reviewer for the helpful comment. As the reviewer suggested, we have elaborated proteomics methods information details in the Materials and Methods section (Lines 362-398). The mass spectrometry data generated from this study have been deposited to the ProteomeXchange Consortium with the identifiers

PXD009441, and we have added it in the revised manuscript also.

Minor concerns

- RPLP1 is a very small protein, only 114 amino acid residues. Assuming the authors used trypsin to digest their samples, this would produce potentially 3 tryptic peptides, depending on how many missed cleavages were used in the database search. Are the mass spectrometry results using protein parsimony algorithms or razor peptides to determine proteins?

Response: Thank you for your insightful comments, and we apologize for any confusion caused by the lack of clarity in our description of Mass methods. As described in the revised manuscript, we used trypsin to digest samples with a maximum allowance of two missed cleavages, and the top3 unique and razor peptides with reporter ion mass tolerance of less than 20ppm were used.

- Could the authors speak to the specificity of their anti-RPLP1 antibody? Perhaps a supplemental figure showing the full Western staining of the cytosol and/or nuclear protein fractions stained with the anti-RPLP1 antibody to show the specificity of binding.

Response: We thank the reviewer for the valuable comments. As you suggested, the source data of CD4⁺ T cells' s nuclear-cytoplasm separation samples stained with anti-RPLP1 antibody was shown as below. The results clearly demonstrated the absence of any non-specific bands in close proximity to RPLP1 (molecular weight \approx 15 kD), thereby establishing the specificity of the anti-RPLP1 antibody.

- Fig 3a states that the F2/R2 binding is not significant between IgG and anti-RPLP1, but the figure does look significant albeit by eye. Could the authors provide the raw

data values used to generate this figure?

Response: Thank you for pointing it out. We have re-generated the ChIP figures with the raw data as follows, and modified it in the revised Fig.3a.

	F1R1			F2R2		
input IgG	25.89	25.23	25.25	24.68	24.89	25.01
input HA	27.48	27.28	27.53	24.17	24.25	24.32
IP IgG	25.66	25.28	25.2	23.58	23.84	23.79
IP HA	26.34	25.1	25.42	23.08	23.2	23.2

Reviewer #2 (Remarks to the Author):

The manuscript by Yang et al identify RPLP1 as a potential factor that regulates HIV-1 LTR through a mechanism that interferes with C/EBPbeta binding. Their data suggests that RPLP1 only inhibits clade B viruses, which have conserved C/EBP sites but do not inhibit transcription of other clades. They perform experiments characterizing the potential role of RPLP1 as a transcriptional repressor for the LTR utilizing cell lines, some primary cells and gain and loss of function experiments. In general, most of the effects are modest being approximately two fold. There are limited experiments that address how RPLP1 activity is regulated in CD4+ T cells. The potentially most novel aspect of the paper is the suggestion that RPLP1 limits HIV-1 transcription in long term non-progressors, however, there are no follow-up experiments that address whether RPLP1 limits transcription in cells from this cohort or influences the susceptibility of these cells to different clades providing little insight if this factor is directly contributing to controller phenotype. Specific comments are below.

Response: We greatly appreciate your time and professional suggestions on this manuscript. To answer these questions, we conducted additional experiments and made extensive revisions. We think that the revised manuscript has adequately addressed your concerns.

1. It would have been interesting to use CD4+ cells from their LTNP to determine if cells were more or less resistant to clade B vs other clades. Similarly, they could potentially knock-down RPLP1 in these cells and see if this altered their ability to support HIV-1 infection and transcription. Such experiments would make a stronger case for a role in limiting expression in the context of the non-progressors.

Response: Thank you for your valuable suggestion. Since long-term non-progressors (LTNPs) are rare individuals to collect, the samples we obtained were exclusively utilized for mass spectrum analysis, thereby precluding subsequent experimentation on CD4+ T cells derived from these individuals. Therefore, to substantiate our findings, we performed additional experiment to provide sufficient and robust evidence supporting our conclusion. As shown in Fig. 1, 6 and Supplementary Fig. 1, we investigated the role of RPLP1 in regulating HIV-1 replication and latency in both cell lines and primary CD4+ T cells, through loss-of-function as well as gain-of-function approaches. Additionally, in the revised manuscript, we also assessed the regulatory

impact of RPLP1 on HIV-1 beyond subtype B. We manipulated RPLP1 levels in CD4+ T cells derived from healthy donors and assessed its influence on HIV-1 subtype C infection. Our findings demonstrate that the replication of subtype C, which is independent of C/EBP β , is not modulated by RPLP1, and we have added these results in revised Supplementary Fig.3g and a description in Lines 198-201.

We appreciate your comprehension and have implemented measures to enhance the interpretation of our findings.

2. The description of the Mass Spec and representation of the volcano plot was confusing. I am assuming this represents samples combined from the three controllers and the two individuals on ART. Also, beyond the Mass Spec, was the overexpression of RPLP1 confirmed? Furthermore, it would have been informative to have more details regarding how RPLP1 expression is regulated in resting and activated CD4+ T cells.

Response: Thank you for your comment. As mentioned above, the rare LTNP samples we obtained were exclusively used for mass spectrum analysis, thereby precluding subsequent experimentation on CD4+ T cells derived from these individuals. Therefore, we adopted a compromise approach to detect the relationship between RPLP1 level and disease progression of AIDS (revised Fig.1f). RPLP1 levels were positively correlated with CD4+T cell counts, indicating that there was a negative correlation between RPLP1 expression and disease progression, implying the clinical significance of its antiviral effect. We also investigated the latency regulatory role of RPLP1 in CD4+ T cells derived from cART-treated study participants living with HIV-1 by silencing the RPLP1 and quantified viral reactivation (Fig. 6i). Besides, the regulation of RPLP1 in resting and activated CD4+ T cells was detected, and found that resting CD4+ T cells obtained from healthy donors exhibited robust expression levels of RPLP1, which were observed to decline upon stimulation with phytohemagglutinin-M (PHA-M) or anti-CD3/CD28 antibodies (revised Fig 6h). The underlying mechanism of the RPLP1 decrease upon the CD4+ T cells activation is another issue that needs to be explored, and we will address it in future investigations.

3. There was limited information concerning other proteins that were identified as being overexpressed or differentially reduce in the controllers. Data for the mass spec are not discussed and do not seem to be included in the manuscript. The only protein identified in the volcano plot is RPLP1.

Response: We discussed other proteins with higher or lower expression. Actually, another protein with significant lower expression were under investigation to elucidate their roles in regulating HIV-1, and we hope to publish the finding in the future.

4. The impact of RPLP1, in general, is modest in all their experiments, being about a two-fold change, regardless of knock-down or over-expression. Might this reflect the reported minimal impact of C/EBP sites in CD4+ T cells? This should be discussed.

Response: Thank you for your suggestion, and we have expanded upon this aspect in the discussion section of revised manuscript.

5. Use of RPLP1 mutations to appreciate the structure and function of this protein in the context of HIV transcription was informative.

Response: Since the structure of RPLP1 has been determined, we investigated the role of RPLP1 in regulating HIV-1 based on it and generated the corresponding deletion mutations for each domain. As shown in Fig.5, the α -helices 2 and 4 of RPLP1 are indispensable for its anti-HIV-1 activity, deletion of which resulted in deficiency of RPLP1 to interact with HIV-1 LTR. Further investigation into potential critical sites that may impede the anti-HIV function of RPLP1 would significantly contribute to our comprehensive understanding of its functionality, and we intend to thoroughly address this matter in future studies.

6. Minor comment, for experiments in Fig. 1, the MFIs for the flow cytometry should also be reported, especially since their data supports, minimal impact on infection and the largest impact on transcription.

Response: According to your suggestions, we calculated the MFI of Fig. 1b, Fig. 1c, and Supplementary Fig. 1a. The results consistently support our findings that RPLP1 levels are inversely correlated with HIV-1 replication. Additionally, we also detected the MFI of Fig. 6c and e, which reflected the role of RPLP1 in regulating HIV-1 latency, and proved that RPLP1 knockdown reactivate latent virus.

7. Minor comment: They indicate as “data not shown” for cell viability. These data should be included in the manuscript.

Response: We detected the viability and growth of cells overexpressing or silenced for RPLP1, and found no significant difference in cell viability between control and RPLP1 knockdown or overexpressing cells, which is consistent with previous reports^{1,2}. We have added these results in revised Fig. 1e and Supplementary Fig. 1d with a description in Lines 110-114. For HIV-1 latent cell lines ACH-2 and C11, we also compared the cell viability of control and RPLP1-sh cells, and added the results in Supplementary Fig. 5 c-d.

Reviewer #3 (Remarks to the Author):

In the manuscript by Zhang and colleagues, the authors describe a surprising role for a ribosomal protein, RPLP1, in HIV-1 infection. The first observation of a role for RPLP1 comes from a proteomics screen comparing CD4 cells from 3 non-progressors and 2 viremic progressors. The authors then describe how the protein is and proviral transcription by binding to the CEBPB binding site at the HIV promoter and thereby competing out the CEBPB transcription factor. The authors lastly test the effect of depleting RPLP1 in 3 study participants living with HIV, and find depletion induces expression of HIV-1.

The results are novel and original. The biochemical work, such as the protein domain

deletions and mutation of TF binding sites, is strong and well-founded. However, some of the conclusions and claims need further data and analysis but when substantiated, the study is highly interesting for a general audience. The current manuscript is somewhat limited, as the manuscript is framed in a clinical context, although it mainly relies on results from cell lines. The authors describe several surprising findings that require further in depth studies to be confirmed.

Response: Thank you very much for your time spent reviewing this manuscript. We appreciate your recognition of the originality of our research, and your positive comments and valuable suggestions have helped us to significantly improve our work. To strengthen and extend the clinical significance of this work, we detected the regulation of RPLP1 in resting and activated CD4+ T cells, the role of RPLP1 in regulating other subtypes of HIV-1, as well as the relationship between RPLP1 levels and disease progression in revised manuscript. We hope that the revised manuscript has thoroughly addressed your concern.

Major comments:

1. Although the initial observation of RPLP1 was done in a small study of CD4 samples of progressors (n=2) and non-progressors (n=3), the strong link that the authors make in the abstract and introduction requires further support. To “show that the ribosomal protein lateral stalk subunit P1 (RPLP1) is expressed at higher levels in LTNPs compared to regular progressors (RPs)” (abstract) the authors should test this in a larger independent cohort.

Also, as both of these contrasting groups (LP vs PR) are extremes, what are the levels of RPLP1 in virally suppressed PLWH, representing the majority of cases?

Response: Thanks a lot for your insightful suggestion. As you recommended, the higher levels of RPLP1 in LTNPs should be examined in an independent cohort to verify its significance, and we have also attempted to perform this assessment. However, the present standard practice in China is to immediately initiate HAART upon diagnosis, making it nearly impossible for us to gather a larger cohort of LTNPs and further conduct the verification. Therefore, to acquire sufficient and robust evidence supporting our conclusion, we conducted additional investigations into the role of RPLP1 in regulating HIV-1 replication and latency in both cell lines and primary cells through loss-of-function and gain-of-function approaches (Fig. 1, 6 and Supplementary Fig. 1). Additionally, in the revised manuscript, we also investigated the regulatory impact of RPLP1 on HIV-1 beyond subtype B in primary CD4+T cells (Supplementary Fig. 3g). We manipulated RPLP1 levels in CD4 cells derived from healthy donors and assessed its influence on HIV-1 subtype C infection. Our findings demonstrate that the replication of subtype C, which is independent of C/EBP β , is not modulated by RPLP1, and we have added these results in Revised Supplementary Fig.3g and a description in Lines 198-201.

Furthermore, we compare the mRNA level of RPLP1 in CD4+ T cells isolated from study participants living with HIV-1. Compared to ones with lower CD4+ cell counts, the mRNA level of RPLP1 in the ones with higher CD4+ cell counts were significantly higher, indicating that there was a negative correlation between RPLP1 expression

and disease progression (revised Fig. 1f).

We deeply appreciate your comprehension and have implemented measures to enhance the interpretation of the presentation.

2. Some work is based on primary CD4 cells (1d and 6h-i), but the overwhelming work is done in cancer cell lines, with reporter HIV. Although the effect they observe is expected to have its strongest effect in monocytes/macrophages, as reinforced in the discussion section, this is never tested in primary material. Some of the unexpected key findings, including the RPLP1 translocation to the nucleus should be tested in primary cells and monocytic cells as well.

Response: We thank the reviewer for the constructive suggestion. We have examined RPLP1 translocation to the nucleus in Hela cells, as well as in primary CD4⁺ cells (Supplementary Fig. 4A). According to the reviewer's suggestion, in revised manuscript we further conducted the immunofluorescence co-localization of RPLP1 in monocytic cells upon HIV-1 infection. Again, RPLP1 was found to translocate into nucleus after THP1 cells were infected with HIV-1, further confirmed the finding that HIV-1 infection induced the re-localization of RPLP1. We have added these results in revised Supplementary Fig. 4a.

3. Several unexpected observations are made and they are jointly linked to the subtype B of the HIV-1. Is the nuclear translocation also only observed in this subtype? As this protein redistribution is key to the mechanism, I would like a more thorough characterization. Does nuclear RPLP1 also occur after stresses, other viruses, or other subtypes of HIV?

Response: We thank the reviewer for the helpful comment. As the reviewer suggested, we investigated the re-localization of RPLP1 upon the infection of other subtypes/groups of HIV-1. The new results showed that all examined subtypes B and C of group M, and groups N, O, and P of HIV-1 induced translocation of RPLP1 to the nucleus. In addition to HIV-1, we also observed re-localization of RPLP1 during VSV infection; however, such translocation was not observed in this case. Notably, RPLP1 didn't inhibit the replication of VSV. Please find these results in revised Supplementary Fig. 4c-d.

These new findings collectively suggest that re-localization alone is necessary but not sufficient for the antiviral activity of RPLP1. Whether nucleic RPLP1 can inhibit the virus replication depends on the viral transcription requirement for C/EBP β . Meanwhile, based on our observation of the differential re-localization of RPLP1 between HIV-1 and VSV infection, we propose that the translocation of RPLP1 to the nucleus may be associated with viral life cycle. The HIV-1 life cycle necessitates nuclear entry, during which RPLP1 may be accompanied into the nucleus; in contrast, VSV infection does not involve nuclear entry and therefore does not result in RPLP1 translocation to the nucleus. We have also added a description and discussion of the new results in Lines 221-227 and Lines 318-324, respectively.

4. The data underlying Fig 6 h-l, where the authors use primary material from PLWH.

Here the data does not match up. They state that 1 million CD4 cells were used, but manage to split the cells for two nucleofections, do a immunoblot but also stimulate and grow cells for another 5 days, which result in massive cell death under normal conditions. In the figure, the observed variation in the p24 levels is close to zero. For reference the variation for comparable method in fig 1d has reasonable variation between measurements. The extremely low levels here also make important to denote the limit of detection. The expected reduction in viability after RPLP1 depletion could lead to this observation.

Given that a minute fraction of CD4 cells in PLWH have intact provirus, these results are highly puzzling, even though they confirm the authors hypothesis.

Also, the HIV subtype of the study participants is not explicitly mentioned, despite its importance. To validate this crucial section, the authors could add study participants with with subtype C to the data underlying fig 6 h-i.

Response: We thank the reviewer for pointing this out. For the latent HIV-1 reactivation assays employed in Fig. 6 h-i, we stimulated control and siRPLP1 nucleofected CD4+ T cells derived from PLWH with PHA, which is a commonly utilized activator in HIV-1 latency-related research, ensuring cell viability and maintaining their physiological state³⁻⁵. Furthermore, in our experiment, on the day of supernatant collection for p24 level measurement, the cells exhibited a healthy state as well. Since that CD4+ T cells were obtained from PLWH who had undergone HAART and exhibited low viral loads, the resulting p24 amount of the culture supernatant was determined to be in concentrations of pg/ml levels. In contrast, the p24 amounts in Fig. 1d were measured from donor derived CD4+ T cells infected with HIV-1 NL4-3, which reached ng/ml levels. This distinction may explain the variation difference between Fig. 6i and Fig. 1d.

Based on the reported epidemiological analysis, it has been determined that the predominant HIV-1 subtype in China is B and circulating recombinant forms (CRF)^{6,7}. Moreover, we have successfully amplified and aligned sequences spanning the Gag region of HIV-1 that infected the study participants living with HIV-1 of Fig. 6i, thereby providing further evidence that individuals included in Fig. 6i were infected with HIV-1 subtype B. The rarity of study participants living with HIV-1 subtype C in China obstructed it to investigate the reactivation ability of RPLP1 on subtype C HIV-1. Therefore, we adopted a compromise approach to investigate the regulatory role of RPLP1 on HIV-1 subtype C in primary CD4+ T cells in both gain of function and loss of function aspects. We collected primary CD4+ cells from three healthy donors, silenced or overexpressed RPLP1 and infected them with subtype C virus. The new results showed that the replication of subtype C was not affected by RPLP1. We have added these results in revised Supplementary Fig. 3g with a description in Lines 198-201.

Minor comments:

1. Please use people first language, e.g. "Study participants living with HIV-1" instead of "HIV-1-infected individuals" or "HIV patients".

Response: We have modified to use people first language in the revised manuscript as the reviewer suggested.

2. How are the cells affected by depletion of a major component of a ribosomal unit. What is the viability of after knock-down? In Fig 1b, there is an almost complete depletion of RPLP1. When over-expressing, does it affect the cellular phenotype? This protein is already highly expressed.

Response: To exclude the possibility that alterations in RPLP1 levels impact cell viability, we conducted a CCK8 assay to compare the viability of cells overexpressing or silenced for RPLP1. The results revealed no significant difference in cell viability between control and RPLP1 knockdown or overexpressing cells, which is consistent with previous reports ^{1,2}, indicating that changes in RPLP1 levels have minimal influence on cell viability. We have added these results in revised Fig. 1e, Supplementary Fig. 1d and Supplementary Fig. 5c-d.

3. Figure 1d. Why the axes so different between empty vectors?

Response: The two experimental controls, VR1012 and scrambled siRNA, were employed in overexpression and knock-out RPLP1 experiments respectively. Although the treatment did not induce any impact on cell viability within each group, certain discrepancies between the controls of the two groups emerged, potentially associated with variations in plasmid and siRNA transfection levels. Furthermore, we conducted a replicate experiment, and it yielded consistent results.

4. Figure 6g, the RPLP1 effect here seems to be derived from a change in beta active levels, not RPLP1. To make quantitative measurements on an immunoblot with these small differences, an alternative method should be used, such as stain-free gels.

Response: Thank you for pointing it out. We have re-performed the experiment on the control and stable RPLP1-sh ACH2 cells, and measured the relative levels of p24. As show in revised Fig. 6g, when β -Actin was comparable among different lanes, RPLP1 was silenced, which accompanied with increase of p24 levels.

5. The introduction and results section would benefit from being more focused and aligned to the results section.

Response: We have modified the introduction and results section in the revised manuscript.

References

- 1 Remacha, M. *et al.* Ribosomal acidic phosphoproteins P1 and P2 are not required for cell viability but regulate the pattern of protein expression in *Saccharomyces cerevisiae*. *Mol Cell Biol* **15**, 4754–4762, doi:10.1128/MCB.15.9.4754 (1995).
- 2 Perucho, L. *et al.* RPLP1, a crucial ribosomal protein for embryonic development of the nervous system. *PLoS One* **9**, e99956, doi:10.1371/journal.pone.0099956 (2014).
- 3 Zhu, J. *et al.* Reactivation of latent HIV-1 by inhibition of BRD4. *Cell Rep*

- 2, 807–816, doi:10.1016/j.celrep.2012.09.008 (2012).
- 4 Sun, W. W. *et al.* SUN2 Modulates HIV-1 Infection and Latency through Association with Lamin A/C To Maintain the Repressive Chromatin. *mBio* **9**, doi:10.1128/mBio.02408-17 (2018).
- 5 Ma, X. *et al.* TRIM28 promotes HIV-1 latency by SUMOylating CDK9 and inhibiting P-TEFb. *Elife* **8**, doi:10.7554/eLife.42426 (2019).
- 6 Yan, M. *et al.* HIV-1 diversity and drug-resistant mutations in infected individuals in Changchun, China. *PLoS One* **9**, e100540, doi:10.1371/journal.pone.0100540 (2014).
- 7 Yuan, R., Cheng, H., Chen, L. S., Zhang, X. & Wang, B. Prevalence of different HIV-1 subtypes in sexual transmission in China: a systematic review and meta-analysis. *Epidemiol Infect* **144**, 2144–2153, doi:10.1017/S0950268816000212 (2016).

REVIEWER COMMENTS

Reviewer #1 (Remarks to the Author):

I thank the authors for the additional information regarding their mass spectrometry experiment, however unfortunately there is still not enough detail provided for me to confidently determine whether the authors have performed appropriate statistical analysis of their mass spectrometry data.

(1) They did provide adequate detail on the acquisition, but there's no description of how they performed a pairwise comparison with uneven replication across the sample groups (3 replicates vs 2 replicates).

(2) Further, the provided dataset identifier PXD009441 maps to a different experiment entitled "Probing the Sensitivity of the Lumos Mass Spectrometer using a Standard Reference Protein in a Complex Background". How were p-values calculated? What statistical test was used to determine differentially expressed proteins?

With that in mind, I am sorry to write that I am not convinced that the authors have sufficiently addressed my major concern regarding their mass spectrometry proteomics in this paper.

Reviewer #2 (Remarks to the Author):

The authors overall were responsive to reviewers' suggestions. A minor suggestion, although it is appreciated the challenges of studying rare LTNP cohorts, with the limited data on controllers, the implied link between this and RPLP1 should be more cautious. The current abstract and introduction still reads as if this paper is about LNTPs rather than mechanisms of RPLP1 regulating HIV transcription.

Reviewer #3 (Remarks to the Author):

The authors have addressed most of my concerns satisfactory and the manuscript has been significantly improved. However, there are still some conclusions that I find lack support. These issues can be resolved by minor textual changes only.

Major point 1: I understand that the LTNPs are rare and expanding the cohort cannot be done. The second statement of the abstract is however very strong and cannot be said to represent all LTNPs based solely on 2-3 samples. Instead of “show”, your data “suggest” the higher levels of RPLP1, or something similar.

The study participants which are referred to as “regular progressors” (RP) have received ART, as specified in the Methods section. This information should be in the main text. Some more information in Supplementary table 1 would be useful, as the RP here do not have the common characteristics of PLWH on ART today. In this cohort the study participants have higher levels of CD4 cells and the viral levels are not suppressed. A few words describing the initial cohort better would improve the manuscript.

Major point 4: In the Methods section, the authors still claim to use 1 million primary CD4 T cells only as starting material for all the experiments. Is this a typo?

The complete lack of variation in fig 6i is not explained. It looks like these are replicate wells from the ELISA rather than independent experiments. Then the statistical tests are not relevant. Fig 6i would be stronger by plotting the change in a single graph of P1-P3 together (maintaining the individual data points), and importantly doing the statistical test between the siNC (P1-P3) and siRPLP1 (P1-P3).

Minor comment 1: It's advisable to use “HIV-negative study participants” instead of “healthy donors”. Also, the word “patients” is still used in the supplemental material.

We would like to thank the reviewers for the thorough comments. In accordance with their valuable suggestions, we have made modifications to both the text and figures, resulting in a significant enhancement of our manuscript. Here, we present a comprehensive point-by-point response to address each of the reviewers' comments (underlined):

REVIEWER COMMENTS

Reviewer #1 (Remarks to the Author):

I thank the authors for the additional information regarding their mass spectrometry experiment, however unfortunately there is still not enough detail provided for me to confidently determine whether the authors have performed appropriate statistical analysis of their mass spectrometry data.

Response: We greatly appreciate your time and professional suggestions on this manuscript. To ensure the accuracy of statistical analysis, we consulted experts specialized in bioinformatics analysis and re-performed statistical analysis on mass spectrometry data, obtained the similar results. For mass spectrometry experiment, we had performed three technical replicates, with the data from run 1 selected as a representative sample for subsequent statistical analysis in the previous version of the manuscript. In this revised version, we have re-analyzed the complete dataset encompassing all three runs, consistently demonstrating that RPLP1 exhibits significantly higher expression in LTNPs. These findings align with our additional experimental evidence suggesting that RPLP1 plays a suppressive role in HIV-1 replication. Detailed information regarding the statistical analysis has been included in the Material and Methods section (Lines 399-412). We hope that the revised manuscript has thoroughly addressed your concern.

(1) They did provide adequate detail on the acquisition, but there's no description of how they performed a pairwise comparison with uneven replication across the sample groups (3 replicates vs 2 replicates).

Response: Thank you so much for your insightful comments. The study participants were categorized into two groups based on disease progression (LTNPs and RPs), and the rarity of study participants obstructed us to collect more samples. While it is ideal for statistical analysis to have an equal number of individuals in each group, variations in sample sizes across groups are deemed acceptable as well. To identify differentially expressed proteins between LTNPs and RPs, we employed the linear models for microarray data (limma)¹, which can be utilized for analyzing uneven replication, and has been applied in the analysis of proteomics data², including articles with high impact factor³ as well as virus-related research⁴. We have added details regarding statistical analysis in Material and Method section (Lines 399-412).

(2) Further, the provided dataset identifier PXD009441 maps to a different experiment entitled "Probing the Sensitivity of the Lumos Mass Spectrometer using a Standard

Reference Protein in a Complex Background". How were p-values calculated? What statistical test was used to determine differentially expressed proteins?

Response: We thank the reviewer for the suggestion, and we apologize for the inadvertent oversight in referencing the dataset identifier number. The mass spectrometry data generated from this study have been deposited to the ProteomeXchange Consortium with the identifiers PXD050294, and we have corrected this (Line 397). As described above, differentially expressed proteins were identified using the limma package, which employed moderated t-test to compare the average normalized protein abundance between LTNPs and RPs, with the criteria that fold change > 2 as well as a *P* value < 0.05.

With that in mind, I am sorry to write that I am not convinced that the authors have sufficiently addressed my major concern regarding their mass spectrometry proteomics in this paper.

Response: Thanks a lot for your valuable feedback. As you recommended, we carefully examined the interpreting of mass data and re-conducted the data analysis. The volcano plot generated from re-analysis results still revealed RPLP1 as the up-regulated gene in LTNPs with the highest fold change. Taken the subsequent functional evidences regarding the regulatory role of RPLP1 on HIV-1 into account, these observations collectively substantiate our conclusion. We now provide a detailed description of the data analysis in the Materials and Methods section (Lines 399-412).

References:

- 1 Ritchie, M. E. *et al.* limma powers differential expression analyses for RNA-sequencing and microarray studies. *Nucleic Acids Res* **43**, e47, doi:10.1093/nar/gkv007 (2015).
- 2 D'Angelo, G. *et al.* Statistical Models for the Analysis of Isobaric Tags Multiplexed Quantitative Proteomics. *J Proteome Res* **16**, 3124-3136, doi:10.1021/acs.jproteome.6b01050 (2017).
- 3 Bech, J. M. *et al.* Proteomic Profiling of Colorectal Adenomas Identifies a Predictive Risk Signature for Development of Metachronous Advanced Colorectal Neoplasia. *Gastroenterology* **165**, 121-132 e125, doi:10.1053/j.gastro.2023.03.208 (2023).
- 4 Borges-Velez, G. *et al.* Zika virus infection of the placenta alters extracellular matrix proteome. *J Mol Histol* **53**, 199-214, doi:10.1007/s10735-021-09994-w (2022).

Reviewer #2 (Remarks to the Author):

The authors overall were responsive to reviewers' suggestions. A minor suggestion, although it is appreciated the challenges of studying rare LTNPs cohorts, with the limited data on controllers, the implied link between this and RPLP1 should be more cautious. The current abstract and introduction still reads as if this paper is about LTNPs rather than mechanisms of RPLP1 regulating HIV transcription.

Response: Thank you very much for your time spent reviewing this manuscript. We have modified the abstract and introduction according to your suggestions, and hope that the revised manuscript has adequately addressed your concern.

Reviewer #3 (Remarks to the Author):

The authors have addressed most of my concerns satisfactory and the manuscript has been significantly improved. However, there are still some conclusions that I find lack support. These issues can be resolved by minor textual changes only.

Response: We are pleased about the positive feedback and made the textual changes as suggested.

Major point 1: I understand that the LTNPs are rare and expanding the cohort cannot be done. The second statement of the abstract is however very strong and cannot be said to represent all LTNPs based solely on 2-3 samples. Instead of “show”, your data “suggest” the higher levels of RPLP1, or something similar.

The study participants which are referred to as “regular progressors” (RP) have received ART, as specified in the Methods section. This information should be in the main text. Some more information in Supplementary table 1 would be useful, as the RP here do not have the common characteristics of PLWH on ART today. In this cohort the study participants have higher levels of CD4 cells and the viral levels are not suppressed. A few words describing the initial cohort better would improve the manuscript.

Response: Thanks for your comment, we modified the abstract (Lines 28-29) and description of RPs (Lines 91-92) as per your recommendations.

Major point 4: In the Methods section, the authors still claim to use 1 million primary CD4 T cells only as starting material for all the experiments. Is this a typo?

The complete lack of variation in fig 6i is not explained. It looks like these are replicate wells from the ELISA rather than independent experiments. Then the statistical tests are not relevant. Fig 6i would be stronger by plotting the change in a single graph of P1-P3 together (maintaining the individual data points), and importantly doing the statistical test between the siNC (P1-P3) and siRPLP1 (P1-P3).

Response: We appreciate your insightful suggestion. The nucleofection of primary CD4 T cells was conducted using the Amaxa™ 4D-Nucleofector platform, and we used one million cells as recommended by the manufacturer. The lack of variation observed in Fig. 6i can be attributed to the replicate wells used in the ELISA assay. In accordance with your comment, we have modified Fig 6i by plotting the change in a single graph of P1-P3 together and performed the statistical test between the siNC (P1-P3) and siRPLP1 (P1-P3).

Minor comment 1: It’s advisable to use “HIV-negative study participants” instead of “healthy donors”. Also, the word “patients” is still used in the supplemental material.

Response: Changed as suggested (Lines 116-117, 198, 207-208, 267-268, 343-344, 355, 519-520, 583, 798-800, 878-879, 946, 969, 1052-1053, 1062 and Supplementary Table 2).

REVIEWER COMMENTS

Reviewer #1 (Remarks to the Author):

I appreciate the authors' efforts to address my concerns, and I admit that they are specialized to the mass spectrometry proteomics.

However, I'm disappointed to note that my two points remain unsatisfactorily addressed.

(1) The additional methods detail is helpful, but simply using the limma R package doesn't inherently deal with the uneven replication. While limma does not require equal variances, specific parameters must be used, e.g. to denote weights, for limma to allow different variabilities for the treatment groups. Perhaps sharing as a supplemental file the R code used to generate the volcano plot, showing how to filter the Proteome Discover data and perform the statistical testing would be helpful to share (especially for transparency and reproducibility)?

(2) Raw data (in this edit, the identifier PXD050294) is not findable in the repository.

#####

Additional comments:

Overall, the code does clarify the analysis (I'd highly recommend the code be shared as supplemental information!), and my main concern about biological replication and data massaging influencing the biological conclusions -- that is presumably address by the other reviewers' expertise looking at the follow-up experiments to validate the mass spectrometry results, and I defer to their judgement on those sections of the paper!

Reviewer #3 (Remarks to the Author):

The authors have addressed my concerns satisfactory.

We would like to thank the reviewer #1 for the comments. As reviewer #1 suggested, we have included the R code as a supplement to the manuscript, specific and detailed methods for the related analysis are also provided with code. Here, we present a comprehensive point-by-point response to address each of the reviewers' comments (underlined):

Reviewer #1 (Remarks to the Author):

I appreciate the authors' efforts to address my concerns, and I admit that they are specialized to the mass spectrometry proteomics.

However, I'm disappointed to note that my two points remain unsatisfactorily addressed.

(1) The additional methods detail is helpful, but simply using the limma R package doesn't inherently deal with the uneven replication. While limma does not require equal variances, specific parameters must be used, e.g. to denote weights, for limma to allow different variabilities for the treatment groups. Perhaps sharing as a supplemental file the R code used to generate the volcano plot, showing how to filter the Proteome Discover data and perform the statistical testing would be helpful to share (especially for transparency and reproducibility)?

Response: Thank you for your comments. As you suggested, we have included the R code with details as a supplement to the manuscript (described in Lines 411-421), and also discussed the technical limitations of the analysis in the discussion section (Lines 330-336).

(2) Raw data (in this edit, the identifier PXD050294) is not findable in the repository.

Response: We have uploaded the raw data of mass in iProx on March 4th, 2024 with the access level as public after acceptance (means that we will open it to public when this MS is accepted). We apologize for the inconvenience due to our caution and unfamiliar with this system. Please verify this data using the provided link (<https://www.iprox.cn/page/PSV023.html?url=1714711122768IU1s>) and key (Fpco), and we will update its access level to public upon our work is accepted. Actually, we would like to update its access level now, but we are afraid of causing some change when we update it, resulting in further misunderstanding.

#####

Additional comments:

Overall, the code does clarify the analysis (I'd highly recommend the code be shared as supplemental information!), and my main concern about biological replication and data massaging influencing the biological conclusions -- that is presumably address by the other reviewers' expertise looking at the follow-up

experiments to validate the mass spectrometry results, and I defer to their judgement on those sections of the paper!

Response: Thanks for your comments. As suggested, we have included the R code as a supplement to the manuscript. Indeed, biological replication and data massaging may influence the conclusions. However, based on valuable insights provided by mass data, subsequent biological experiments further supported the important role of RPLP1 in regulating HIV-1, which is the major part of our work. Taken the analysis of mass data and lots of biological experiments together, we concluded that RPLP1 inhibits HIV-1 replication by occupying the C/EBP β binding sites in the viral LTR.